

# Post-hoc reweighting of hadron production in the Lund string model

Benoît Assi[1*], Christan Bierlich[2†], Philip Ilten[1‡], Tony Menzo[1○], Stephen Mrenna[1,3§],
Manuel Szewc[1,4¶], Michael K. Wilkinson[1‖], Ahmed Youssef[1♠] and Jure Zupan[1⋈]

**1** Department of Physics, University of Cincinnati, Cincinnati, Ohio 45221, USA
**2** Department of Physics, Lund University, Box 118, SE-221 00 Lund, Sweden
**3** Computational Science and AI Directorate, Fermilab, Batavia, Illinois, USA
**4** International Center for Advanced Studies (ICAS), ICIFI and ECyT-UNSAM,
25 de Mayo y Francia, (1650) San Martín, Buenos Aires, Argentina

★ assibt@ucmail.uc.edu , † christian.bierlich@fysik.lu.se , ‡ philten@cern.ch ,
○ menzoad@mail.uc.edu , § mrenna@fnal.gov , ¶ szewcml@ucmail.uc.edu ,
‖ michael.wilkinson@uc.edu , ♠ youssead@ucmail.uc.edu , ⋈ zupanje@ucmail.uc.edu

## Abstract

We present a method for reweighting flavor selection in the Lund string fragmentation
model. This is the process of calculating and applying event weights enabling fast and
exact variation of hadronization parameters on pre-generated event samples. The procedure is *post hoc*, requiring only a small amount of additional information stored per
event, and allowing for efficient estimation of hadronization uncertainties without repeated simulation. Weight expressions are derived from the hadronization algorithm
itself, and validated against direct simulation for a wide range of observables and parameter shifts. The hadronization algorithm can be viewed as a hierarchical Markov
process with stochastic rejections, a structure common to many complex simulations
outside of high-energy physics. This perspective makes the method modular, extensible,
and potentially transferable to other domains. We demonstrate the approach in PYTHIA
8, including both coverage considerations and timing benefits. For the purpose of this paper, our goal is to develop and demonstrate the the formalism, and we therefore exclude
several model variations for baryon production (popcorn model, junction production)
needed for proton collisions. These will be the topic of a future paper.

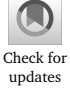

# 1 Introduction

Hadronization models describe the dynamics of the transition from a partonic to a hadronic final state in simulations of high-energy collisions of leptons, hadrons, and heavy nuclei. This simulation step is essential, since hadrons, and not partons, are the objects directly accessible to detector experiments [1]. In general purpose Monte-Carlo event generators such as PYTHIA 8 [2], HERWIG 7 [3], and SHERPA [4], hadronization is modeled using the Lund string model [5] and the cluster model [6]. These models have served as the standard for decades, playing a critical role in shaping our understanding of various phenomena, many of which are not directly related to the hadronization mechanism itself [7]. In the past decade, for instance, the unexpected discovery of quark–gluon–plasma–like behavior in proton–proton collisions [8,9] has sparked renewed interest in hadronization models [10–17] as a means of providing an alternative explanation of these emerging phenomena.

In all these cases, reliable parameter estimation—along with a proper assessment of uncertainties—is a crucial aspect. Recently, we introduced a modified accept-reject Monte-Carlo sampling algorithm [18], which enables a less computationally intensive exploration of how variations in hadronization parameters affect simulated observables by reweighting already-generated events. The method was initially limited to parameters in the so-called Lund symmetric fragmentation function, which controls the distribution of momenta in string breaks. In this paper, we extend this approach by introducing a novel reweighting method for the flavor selection part of the PYTHIA 8 hadronization algorithm.

The *event weight* is a central concept in both methods. A Monte-Carlo simulated collision event is, roughly speaking, constructed by selecting its different attributes, and thus also the attributes of outgoing particles, from probability distributions describing the underlying physics. It is a sequential process, where the selection from one distribution feeds the next one, *e.g.*, when selecting an incoming parton from a parton density function (PDF), which then participates in a hard scattering. Naively, an imagined perfect model, performing a correct selection of all event attributes, will result in simulated events appearing in the same proportion as

provided by nature, *i.e.*, with event weights of unity. One can, however, choose to oversample or undersample particular processes, or parts of phase space, in a controlled way. The over-sampling/undersampling can then be corrected with a correspondingly smaller/larger event weight. This type of reweighting scheme has been implemented in various parts of the collision simulation process for some time [19–24], but it was only recently implemented for the Lund symmetric fragmentation function [18].

In this paper we extend the oversampling/undersampling procedure, corrected by event weights, to flavor selection in the hadronization procedure. As proof-of-principle, we fully validate this procedure for string systems without junctions or baryon production via the popcorn mechanism. Both these model variations are needed for realistic modeling of proton collisions, and will be the topic of a future paper. In the process we introduce two different computational methods for obtaining the weights for flavor parameter variations, which we term the *analytic* and the *stochastic* prescriptions. Both prescriptions are designed to reweight distributions of physical observables from one point in parameter space to another, but *differ in how they account for the rejected proposals* for produced particles. These rejected proposals, although not affecting explicitly the observables of interest, need to be taken into account in order to correctly compute the probability of a physical observable given a set of parameters.

In the *analytic prescription*, the rejected proposals are explicitly integrated out to produce closed-form equations for the weights. The weights then depend on the parameters being varied, and the different filter efficiencies involved in the simulation chain. The explicit, analytic integration is what gives the prescription its name. This method is particularly useful for illustrating the logic behind how the flavor weights can be constructed. However, a full version of the analytic method is tedious to derive in practice. Thus, we also developed a *stochastic* prescription that, as long as the efficiencies remain unchanged as we vary the parameters of interest, produces weights that are equivalent to the analytic method for an ensemble of events, while the weights do differ on an event-by-event basis. In the stochastic method the full rejection history is retained along with the accepted states. The name of the stochastic prescription reflects the fact that we do not perform an analytic sum but instead a summation over the actual sampling history. The stochastic prescription can thus be viewed as a way to make a Monte Carlo estimate of the full summation.

The paper is structured as follows. In section 2 we describe the process of forming hadrons through the Lund string model and how weights can be generated from flavor parameter variations, in increasing steps of complexity. In section 3 we test the validity of our implemented method, including timing tests. Section 4 presents conclusions and possible future directions. Further details are provided in the appendices: appendix A describes how reweighting procedures work in simulations that contain a filtering step, and appendix B contains the derivation of the Clebsch–Gordan weights used in the flavor model.

## 2 Hadron formation through string breaking

The Lund string model [5] is based on identifying the confining color field between a colored and an anti-colored object, *e.g.*, a $q\bar{q}$ pair, with a "massless relativistic string" having string tension $\kappa \approx 1$ GeV/fm. As the confined objects move apart, kinetic energy is transferred into the potential energy of the string; once it is energetically favorable, the string breaks up into smaller pieces, associated with hadrons. This is the basic mechanism whereby (unobservable) colored partonic states in the PYTHIA 8 Monte-Carlo event generator [2] undergo a transition to (observable) hadronic final states. We here review the parts of the model particularly relevant for flavor selection; for more details, we refer the reader to ref. [2] and the references therein.

String breaking occurs via a tunneling process, in which a quark and anti-quark pair tunnels through a classically forbidden region of size $2\sqrt{m^2 + p_\perp^2}/\kappa$ (with $m$ and $p_\perp$ the mass and the transverse momentum of the quark and the anti-quark in the pair, respectively) before they can come on-shell [25]. The governing equation for the tunneling probability $\mathcal{P}$ is

$$\frac{\mathrm{d}\mathcal{P}}{\mathrm{d}^2 p_\perp} \propto \exp(-\pi(m^2 + p_\perp^2)/\kappa) = \exp(-\pi m^2/\kappa)\exp(-\pi p_\perp^2/\kappa). \tag{1}$$

In principle, the relative production rates of light flavored quarks can be estimated directly from eq. (1) by inserting the values for quark masses. However, it is not obvious which definition of the quark masses to use: current or constituent mass or something else [2]. In the Lund approach, the difference between the $u$ and $d$ quark masses is ignored, while the suppression of the generation of $s$-quarks relative to $u$ and $d$ quarks is encoded in a parameter ($\rho$) that is fit ("tuned") to data. The simplest model of baryon production[1] is obtained by allowing diquark–anti-diquark ($\mathcal{Q}\bar{\mathcal{Q}}$) breakups by the same mechanism [29]. The resulting procedure involves four basic parameters for string breaks with default values set to those found in ref. [30], each restricted to values between 0 and 1 inclusive:[2]

$\rho$: The suppression of $s\bar{s}$ string breaks relative to $u\bar{u}$ or $d\bar{d}$. The PYTHIA 8 name and default value for this parameter is `StringFlav:probStoUD = 0.217`.

$\xi$: The overall suppression of a diquark–anti-diquark splitting relative to a $q\bar{q}$ one. The PYTHIA 8 name and default value for this parameter is `StringFlav:probQQtoQ = 0.081`.

$x$: Suppression of diquarks with strange quark content *in addition* to the normal suppression from $\rho$. The PYTHIA 8 name and default value for this parameter is `StringFlav:probSQtoQQ = 0.915`.

$y$: Suppression of spin-1 diquarks relative to spin-0 ones, in addition to the factor of 3 due to spin counting. The PYTHIA 8 name and default value for this parameter is `StringFlav:probQQ1toQQ0 = 0.0275`.

The full process is illustrated in fig. 1. The string breaking accounts for the first part of a two-step process, which divides the string down into a number of smaller string pieces, each with flavor (quark or diquark) input from two neighboring breakup vertices.

The hadron species are not fully determined by the string breaks. For example, neighboring string breaks of $u\bar{u}$, $d\bar{d}$ type result in a $u\bar{d}$ pair connected by a string piece in the middle. This pair can become a $\pi^+$ or a $\rho^+$ meson, with probabilities determined by yet another parameter, called $y_{ud}$, governing the relative production ratio of vector (spin-1) to pseudoscalar (spin-0) mesons for $u$ and $d$ types.[3] A similar parameter governing the relative production ratio of vector to pseudoscalar mesons for $s$ types is here denoted $y_s$.[4]

The $\eta$ and $\eta'$ mesons add a further complication to the algorithm. These particles are overproduced, and must be suppressed to match data. This suppressed production is achieved in the MC simulation through individual tuneable parameters, which we denote in this paper

---

[1]Several more involved models for baryon production, the popcorn model [26] and the junction model [27,28], are also implemented in PYTHIA 8 and can be optionally activated by a user. These are beyond the scope of this paper.

[2]The presented parameterization is not the only possible one. Another one based on hyper-fine splitting effects arising from the mass differences between the light quarks has been developed [16], and one could imagine others. In this paper we have limited ourselves to the simplest model choice.

[3]The PYTHIA 8 name and default value for the parameter $y_{ud}$ is `StringFlav:mesonUDvector = 0.5`. This default makes it twice as likely to sample, *e.g.*, a $\pi^+$ with respect to a $\rho^+$. This parameter includes spin counting, hence its maximal possible value is set to 3.

[4]The PYTHIA 8 name and default value for the parameter $y_s$ is `StringFlav:mesonSvector = 0.55`.

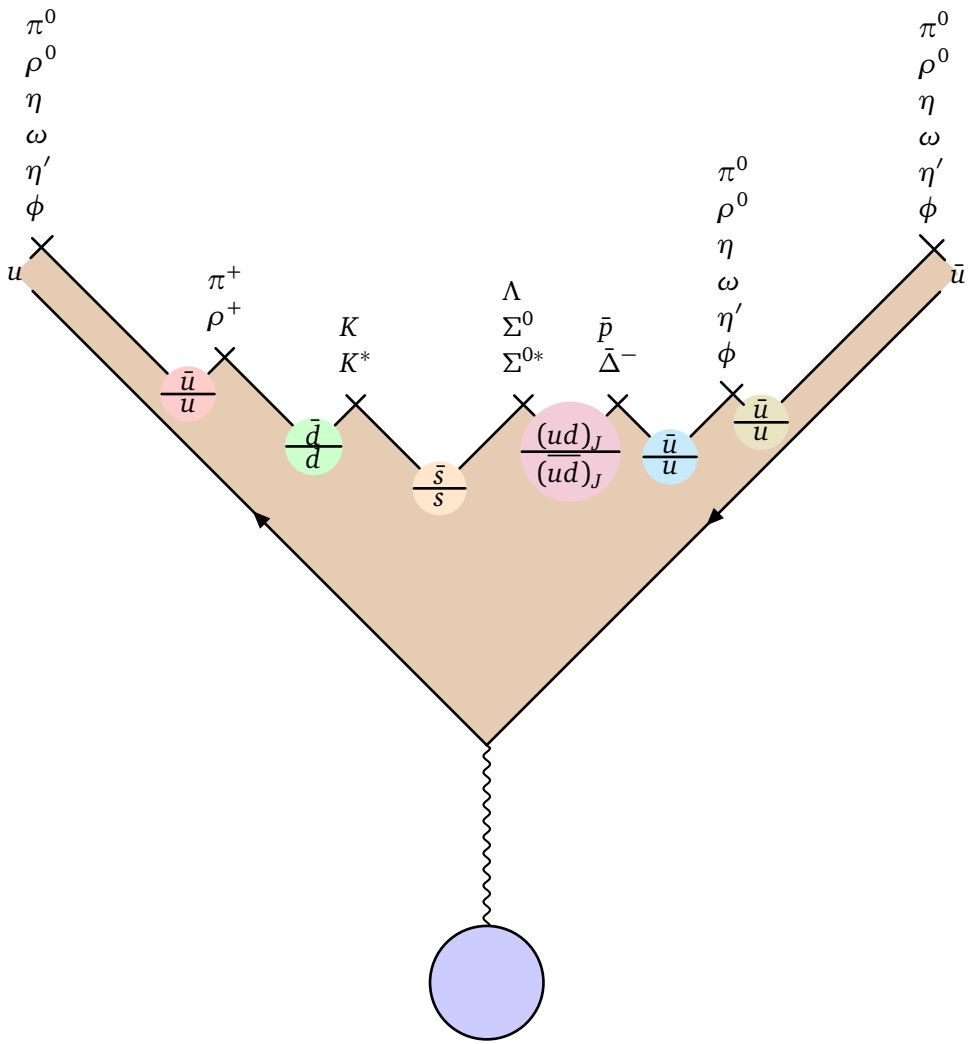

Figure 1: An illustration of the multi-step procedure required to go from strings to hadrons. Reading from bottom to top, a parton-level interaction (blue circle) produces a color singlet quark–antiquark system (lines with arrows). Additional (di)quark-anti(di)quark pairs (colored circles) are produced in the color field (brown area) of the original pair through a vacuum tunneling mechanism. The string is broken into smaller pieces by intersecting combinations of flavored partons. The string pieces will in turn be mapped to hadrons, with species selection steered by Clebsch–Gordan weights and model parameters. Note that the $u\bar{u}$ piece in the upper left part of the figure is allowed to form a $\phi$-meson. With default PYTHIA 8 parameters (almost ideal mixing in the vector meson nonet) this happens for less than 1% of the $u\bar{u}$ string pieces.

as $\epsilon_\eta$ and $\epsilon_{\eta'}$, respectively. They are both suppression probabilities, "filters", meaning that for the value $\epsilon_\eta = 0$ no $\eta$ mesons get produced, while for $\epsilon_\eta = 1$ no rejection takes place, and similarly for $\eta'$.[5] This adds a filtering step to the algorithm, where an $\eta$ or $\eta'$ meson may first get chosen, but then gets rejected by the filter, after which the string breaking procedure must start over again.

---

[5]The PYTHIA 8 names and default value for the parameters $\epsilon_\eta$ and $\epsilon_{\eta'}$ are `StringFlav:etaSup = 0.60` and `StringFlav:etaPrimeSup = 0.12`, respectively.

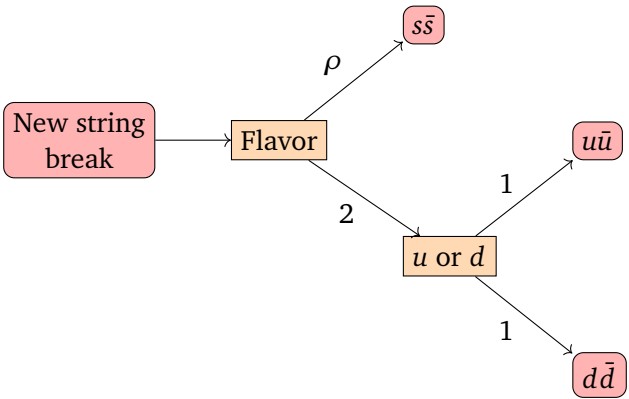

Figure 2: Illustration of the decision flow in the simple case of string breaks without diquarks. There are three possibilities, $u\bar{u}$, $d\bar{d}$, and $s\bar{s}$, the latter with a probability $\rho/(2+\rho)$, and the two former with a probability of $1/2$, given that no $s\bar{s}$ pair was produced. All probabilities can be read off directly from the decision flow chart.

Baryon production from a string piece with a quark and a diquark at the ends, is similar in principle. The string break probabilities follow a similar selection scheme as for quark–antiquark breaks, and the final hadron can be rejected by a filter. In this case, the filter enforces the SU(6) spin-flavor symmetry. That is, if all three light quarks were produced in equal amounts, the resulting hadrons should follow SU(6) spin-flavor symmetry. This is ensured using SU(6) Clebsch–Gordan coefficients for individual quark–diquark combinations, making up the baryons in the octet and the decuplet. See section 2.5 and appendix B for further details. As before, cases may arise where no hadron is chosen, and the string breaking procedure must start over again.

For the purpose of this paper, it is useful to view the string hadronization process as a Markov chain process: each string break corresponds to a probabilistic decision, conditioned on the previous state, and feeds into the next. Rejections and filters act as intermediate selection steps but preserve the sequential structure as long as the rejected decisions are still accounted for. This perspective naturally lends itself to factorized weight calculations, as we develop below. In the following sub-sections we will give a step-by-step explanation of how a reweighting strategy can be formulated as a modification to the existing string break algorithm. In section 2.1 we illustrate the procedure in the simplest version of the problem, excluding diquarks and neglecting rejection at the hadron formation level. In section 2.2 we include rejection at the hadron formation level for the simple version, and in section 2.3 we build on the simple example to introduce the final formalism of stochastic weights. In section 2.4 we introduce weights for diquark breaks, and finally in section 2.5 we introduce rejection at the hadron formation level for baryon production as well.

## 2.1 String break weights: A simple example

First, let us consider a simple case where strings are allowed to break only into $q\bar{q}$ pairs of the lightest quark flavors, $u$, $d$, or $s$. That is, in this first example we do not allow a creation of diquarks, and thus no baryons. For this case a very simple decision tree can be drawn, as shown in fig. 2.

As shown in the decision tree and already described in the introductory part of section 2, an $s\bar{s}$ break is suppressed by a factor $\rho$ with respect to $u\bar{u}$ or $d\bar{d}$ breaks, giving a probability of $p = \rho/(2+\rho)$ for an $s\bar{s}$ break. The probability that a string with a total of $N$ breaks produces

exactly $n$ $s\bar{s}$ breaks is thus given by

$$\mathcal{P}_{ns}(\rho) = \binom{N}{n} p^n (1-p)^{N-n}, \tag{2}$$

where $\binom{N}{n}$ is the binomial coefficient. This is the result of independently sampling $N$ indistinguishable string breaks, where each string break is either an $s\bar{s}$ break (with probability $p$) or not (with probability $1-p$). The *event weight*, which is our main object of interest, can thus be determined by probabilities of the type computed as in eq. (2). Schematically, the event weight after a change of parameters is simply

$$w \equiv \frac{\mathcal{P}_{\text{breaks}}(\text{params.}')}{\mathcal{P}_{\text{breaks}}(\text{params.})} w_{\text{event}}, \tag{3}$$

where $w_{\text{event}}$ is the original event weight which we assume to be unity in the following. These weights can be straightforwardly combined with weights originating from other parameter variations, such as those considered in ref. [18], by multiplying them together. In this simplified example, a change $\rho \mapsto \rho'$ gives $p \mapsto p' = \rho'/(2+\rho')$, and the weight of the event changes from unity to[6]

$$w = \frac{\mathcal{P}_{ns}(\rho')}{\mathcal{P}_{ns}(\rho)} = \left(\frac{p'}{p}\right)^n \left(\frac{1-p'}{1-p}\right)^{N-n}. \tag{5}$$

Here, we have considered the weight for a string with a fixed number of breaks, $N$. This will always be the case; eq. (5) is defined for a fixed $N$. This is counter to the intuition one would have as a user of Monte Carlo event generators. For an ensemble of events, each with an in principle different number of total string breaks $N$, parameter variations can effect events of varying $N$ differently, modifying $\langle N \rangle$ and thus shifting particle multiplicity distributions. For example, increasing $\rho$ can decrease $\langle N \rangle$. A natural question then is how can eq. (5) account for these multiplicity shifts, if $w$ is only defined for a fixed $N$. The critical point is that while $N$ does not change for a varied parameter, $w$ does. In this example, increasing $\rho$ would decrease the weights for events with high $N$, and increase the weights for events with low $N$. In this way, $N$ itself does not change for any single event or chain, even though the weighted $\langle N \rangle$ and multiplicity distributions do. Of course, these reweighted distributions only remain valid when there is support from the baseline distribution.

The weight of eq. (5) is the basic component of both the analytic and stochastic methods we will describe. The main benefit of this weight is that it can be calculated for arbitrary variations of the $\rho$ parameter—it acts as a local Jacobian between the original and target parameterizations of the string break probability space—as long as $N$ and $n$ are retained through the event generation process, *i.e.*, a technical implementation only requires storing two additional integers per event ($N$ and $n$). In effect, the stored per-event integers act as a minimal sufficient statistic for computing reweighting factors under arbitrary changes to the flavor parameters. In practice, however, the hadronic composition is not fully determined by determining the amount of $s\bar{s}$ breaks in an event. We handle this complication next.

## 2.2 Rejection weights: Making hadrons out of simple string breaks

As alluded to in the beginning of section 2, a fraction of produced $\eta$ and $\eta'$ mesons are filtered between the string-break step, and the final production of the hadron. The filtering is dictated

---

[6]While eq. (5) can be simplified to

$$w = \left(\frac{\rho'}{\rho}\right)^n \left(\frac{2+\rho}{2+\rho'}\right)^N, \tag{4}$$

the expression in eq. (5) is more easily generalized to diquark production.

by the parameters $\epsilon_\eta$ and $\epsilon_{\eta'}$, respectively. The situation is further complicated by the existence of mixing, in particular in the pseudoscalar sector, and to a smaller extent the vector-meson sector.

The inclusion of filtering, even in the simplified example, thus, quickly becomes complicated. The full algorithm for producing a meson from a string break is as follows when starting with $s$-quark as the end-point. This means that the string break preceding this break, was $s\bar{s}$. This case turns out simpler than starting with $u$ or $d$ end-point quarks. Rejections in the following are always at the level of a single break, so the initial $s$-quark is never rejected, reflecting the Markovian nature of the process. As such, we can keep the initial $s$-quark as a starting point:

1. Select the string break flavor, $\bar{s}$ with weight $\rho$ and $\bar{u}$ or $\bar{d}$ with a combined weight 2.

2. If a $\bar{u}$ or $\bar{d}$ is chosen, a kaon is produced. Further logic determines its spin as well as mixing, but there are no rejections along this algorithm branch.

3. If an $\bar{s}$ is chosen, spin must be selected to choose whether a vector meson or a pseudoscalar meson is created. This is determined by the $y_s$ parameter defined in section 1.

4. If a vector meson is chosen, further logic determines mixing. This has almost negligible effect for $s\bar{s}$ vector states. There are no rejections along this algorithm branch.

5. If a pseudoscalar meson is selected, an $\eta'$ is chosen with probability $\sin^2(\alpha)$ and $\eta$ with probability $\cos^2(\alpha)$, where $\alpha$ is an angle needed to specify the probability of projecting on to a given meson state.[7]

6. If an $\eta'$ was chosen, accept it with probability $\epsilon_{\eta'}$. If rejected, return back to step 1.

7. If an $\eta$ was chosen, accept it with probability $\epsilon_\eta$. If rejected, return back to step 1.

In order to have a more pedagogical explanation of the reweighting algorithm, including filtering, let us make a few simplifications to the above algorithm: let us allow only production of pseudoscalars, and, furthermore, out of $\eta, \eta'$ allow only production of $\eta$ mesons. This gives the simpler procedure shown in fig. 3, where the observables of interest are only functions of the number of accepted $s\bar{s}$ breaks and the number of $u\bar{u}$ and $d\bar{d}$ breaks with no knowledge of any rejected $s\bar{s}$ breaks. We want to make clear that this simplified procedure is only intended as a pedagogical toy to introduce the formalism, no physics conclusions should be drawn from it. In the full implementation, which we will present later, all production modes are accounted for. In this simplified algorithm, the probability to break into $s\bar{s}$, is now effectively reduced by the presence of a rejection to

$$p_{s\bar{s}|s,\text{eff}} = \frac{p_{s\bar{s}|s}\epsilon_\eta}{(1-p_{s\bar{s}|s}) + p_{s\bar{s}|s}\epsilon_\eta} = \frac{\rho\epsilon_\eta}{2+\rho\epsilon_\eta}\,, \tag{6}$$

where the last expression is derived by replacing $p_{s\bar{s}|s} = \frac{\rho}{2+\rho}$. In appendix A we derive the more general case where a break can produce one of $K$ states with probabilities $\{p_k\}_{k=1}^K$ each of which has a filter efficiency $\{\epsilon_k\}_{k=1}^K$. In that case, the effective probability becomes

$$p_{k,\text{eff}} = \frac{p_k\epsilon_k}{\sum_{k=1}^K p_k\epsilon_k}\,, \tag{7}$$

with $\sum_k p_{k,\text{eff}} = 1$ by definition. We can recover eq. (6) by setting $K = 2$, identifying $k = 1$ with $s\bar{s}|s$ (where $p_{s\bar{s}|s} = \frac{\rho}{2+\rho}$ and $\epsilon_{s\bar{s}|s} = \epsilon_\eta$) and $k = 2$ with $s\bar{u}+s\bar{d}|s$ (where $p_{s\bar{u}+s\bar{d}|s} = 1-p_{s\bar{s}|s} = \frac{2}{2+\rho}$ and $\epsilon_{s\bar{u}+s\bar{d}|s} = 1$).

---

[7]The angle $\alpha$ is related to the normal mixing angle $\alpha = \theta + 54.7$ degrees, see the review of the quark model in ref. [31] for terminology and details. In PYTHIA 8 this parameter is called `StringFlav:thetaPS` and has a default value of -15 degrees.

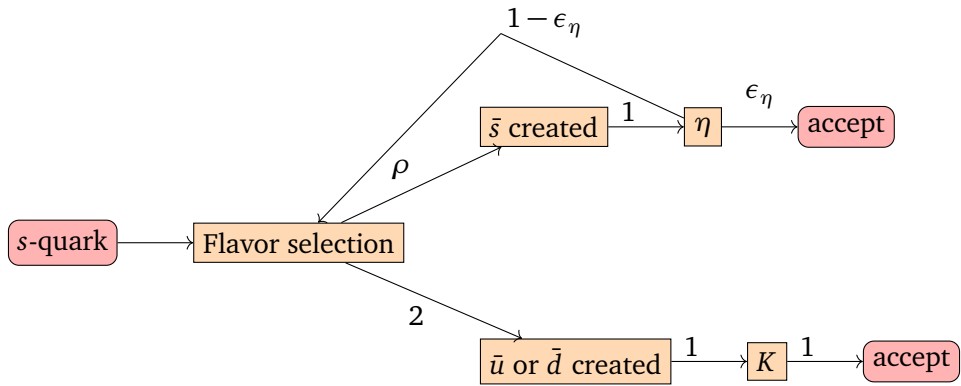

Figure 3: Flowchart depicting the introduction of rejection/filtering in a simplified example beginning with an $s$-quark end-point. In this toy example, vector mesons and the $\eta'$ meson are not included, so a string break following an $s$-quark can only result in an $\eta$ (for $s\bar{s}$ breaks) or kaons (for $u\bar{u}$ or $d\bar{d}$ breaks). Because of this simplification, this example does not conserve isospin. However, the full reweighting implementation of this paper does.

In section 2.1 we introduced the reweighting of simple string breaks as a consequence of the ability to express the number of $s\bar{s}$ breaks $n$ as a binomially distributed variable. This is possible since the simple string break does not depend on the previous step (*i.e.*, it is a Bernoulli trial). The above discussion, and eq. (6) in particular, show that even in the presence of filtering, simple reweighting is still possible; one just needs to replace the probability $p$ in eq. (5) with an appropriately chosen effective probability $p_{\text{eff}}$. We emphasize that this reweighting procedure does not make any changes to the modeled events, only their weights. As such, simulated events follow the complete underlying physics of the implemented model (conservation laws *etc.*). Analytic expressions for reweighting factors, however, become very involved very quickly with effective probabilities introduced for many different kinds of rejections. For instance, the full effective probability to produce an $s\bar{s}$-pair, starting from an $s$-quark, $p_{s\bar{s}|s,\text{eff}}$, correcting for all rejections in the algorithm outlined above, but still omitting baryon production, is given by,

$$
\begin{aligned}
p_{s\bar{s}|s,\text{eff}} &= p_{s\bar{s}_1|s,\text{eff}} + p_{s\bar{s}_0|s,\text{eff}} \\
&= p_{s\bar{s}_1|s,\text{eff}} + p_{\eta|s,\text{eff}} + p_{\eta'|s,\text{eff}} \\
&= \frac{p_{s\bar{s}_1|s}\,\epsilon_{s\bar{s}_1}}{\sum_k p_{k|s}\epsilon_k} + \frac{p_{\eta|s}\,\epsilon_\eta}{\sum_k p_{k|s}\epsilon_k} + \frac{p_{\eta'|s}\,\epsilon_{\eta'}}{\sum_k p_{k|s}\epsilon_k} \\
&= \frac{\rho[y_s + (\epsilon_\eta \cos^2(\alpha) + p_{\eta'}\sin^2(\alpha))(1-y_s)]}{2 + \rho[y_s + (\epsilon_\eta \cos^2(\alpha) + \epsilon_{\eta'}\sin^2(\alpha))(1-y_s)]} \\
&= \frac{\rho_{\text{eff}}}{2 + \rho_{\text{eff}}}, \text{ where } \rho_{\text{eff}} = \rho[y_s + (\epsilon_\eta \cos^2(\alpha) + \epsilon_{\eta'}\sin^2(\alpha))(1-y_s)].
\end{aligned}
\tag{8}
$$

Above, the $p_{s\bar{s}_1,\eta,\eta'|s}$ are the probabilities of sampling, given an $s$ break, a vector $s\bar{s}$ pair, $\eta$ meson, or $\eta'$ meson, respectively. Filter efficiencies lead to an additional suppression, as intended, with $\rho_{\text{eff}} \leq \rho$. Note that eq. (6) is recovered when taking $y_s = 0$ and $\alpha = 0$.

## 2.3 From analytic to stochastic weights

Rather than calculating increasingly complex effective probabilities for each filter, we can instead keep track of rejected and accepted proposals directly. This leads to a *stochastic weight*

*formulation* that bypasses the need for analytic expressions of $p_{\text{eff}}$. For illustrative purposes, we still limit the discussion only to variations in the $\rho$ parameter and keep the number of filter efficiencies fixed. Note, though, that the results obtained below generalize rather straightforwardly; the equivalence between analytic and stochastic weights in the general case is shown in appendix A.3.

The stochastic weight is defined in a manner similar in spirit to the modified accept-reject algorithm introduced for light-cone momentum fraction sampling in ref. [18]. For an accepted string break which has $N_R$ associated rejected breaks we have:

$$w^{\text{break}}_{\text{stochastic}} = \left(\prod_{n=1}^{N_R} w_n\right) w_A = \left(\prod_{n=1}^{N_R} \frac{p'_n}{p_n}\right) \frac{p'_A}{p_A}. \tag{9}$$

Here, $n = \{1, \ldots, N_R\}$ labels the $N_R$ rejected hadron candidates, and $A$ denotes the accepted one. Each term $w_i = p'/p$ reflects the ratio between the target and sampled probabilities for that break.[8]

Let us illustrate this with an example. Consider a simulation sequence where an $\eta$ meson is accepted after having rejected one $\eta$ and one $\eta'$ meson. The stochastic weight for this sequence is

$$w^{\text{break}}_{\text{stochastic}} = w_\eta^2 w_{\eta'} = \left(\frac{(1-y'_s)\rho'\cos^2(\alpha')}{(1-y_s)\rho\cos^2(\alpha)} \frac{(2+\rho)}{(2+\rho')}\right)^2 \left(\frac{(1-y'_s)\rho'\sin^2(\alpha')}{(1-y_s)\rho\sin^2(\alpha)} \frac{(2+\rho)}{(2+\rho')}\right), \tag{10}$$

which for unchanged $y_s$ and $\alpha$ reduces to

$$w^{\text{break}}_{\text{stochastic}} = \left(\frac{\rho'}{\rho} \frac{(2+\rho)}{(2+\rho')}\right)^3, \tag{11}$$

*i.e.*, recovering the form of eq. (4). The key observation is that for any physical observable, reweighting using $w^{\text{break}}_{\text{stochastic}}$ yields equivalent results to incorporating the filter explicitly in terms of $p_{\text{eff}}$. The reason is that physical observables depend only on accepted fragmentations.

Let us write out this statement in terms of expectation values, to show the equivalence explicitly. Let the analytic weights be

$$w^{\text{break}}_{\text{analytic}} = w_{A,\text{eff}} = \frac{p'_{A,\text{eff}}}{p_{A,\text{eff}}}, \tag{12}$$

where the subscript $A$ refers to the accepted hadron, and $p_{\text{eff}}$ is the analytic effective probability to accept $A$, taking into account also the possibility of rejections.

In the stochastic formulation, the full weight for a single string break includes the marginalized contribution from all rejected candidates prior to acceptance. To compute the expectation value of an observable $\mathcal{O}$ that depends only on the accepted hadron, we average over both the accepted candidate $A$ and the entire sequence of rejections. This gives:

$$\mathbb{E}_{A,R}\left[w^{\text{break}}_{\text{stochastic}} \mathcal{O}(\text{Accepted})\right] = \sum_{A=1}^{K} \mathcal{O}(\text{Accepted}) w_A p_A \epsilon_A \sum_{N_R=0}^{\infty} \prod_{n=1}^{N_R} \left(\sum_{k_n=1}^{K} w_{k_n} p_{k_n}(1-\epsilon_{k_n})\right), \tag{13}$$

where we explicitly denoted that the observable only depends on the accepted hadrons, $\mathcal{O}(\text{Accepted})$, and that the expectation value is to be calculated over all accepted and rejected instances. The first sum in eq. (13) runs over accepted hadron types $A$, while the second sum

---

[8]Obtained from eq. (5) by setting $n, N = 1$ and considering the appropriate sampled and target probabilities for the string breaks $p_{n,A}$ and $p'_{n,A}$, respectively.

and the product account for all possible chains of $N_R$ rejected hadron candidates. For each rejection, the stochastic weight $w_{k_n}$ captures the reweighting of the proposal, and the factor $(1 - \epsilon_{k_n})$ reflects the rejection probability.

Assuming proper normalization of both the sampling and target distributions, we can rewrite the above in terms of the target distribution,

$$\mathbb{E}_{A,R}\left[w_{\text{stochastic}}^{\text{break}}\mathcal{O}(\text{Accepted})\right] = \sum_{A=1}^{K}\mathcal{O}(\text{Accepted})p_A'\epsilon_A\sum_{N_R=0}^{\infty}\prod_{n=1}^{N_R}\left(\sum_{k_n=1}^{K}p_{k_n}'(1-\epsilon_{k_n})\right). \quad (14)$$

It is straightforward to show that this matches the structure of the analytic weight construction, eq. (12), with the effective probabilities given in eq. (7). To do so, we notice that performing the sum over all possible hadron types in each rejected hadron allows us to transform the probability $p_A'$ into the effective probability $p_{A,\text{eff}}'$. Explicitly, using $\sum_{k_n}p_{k_n}' = 1$, we can rewrite the second line in eq. (14) as

$$\begin{aligned}
&\sum_{A=1}^{K}\mathcal{O}(\text{Accepted})p_A'\epsilon_A\sum_{N_R=0}^{\infty}\left(1-\sum_{k_n=1}^{K}p_{k_n}'\epsilon_{k_n}\right)^{N_R} \\
&= \sum_{A=1}^{K}\mathcal{O}(\text{Accepted})\frac{p_A'\epsilon_A}{\sum_{k_n=1}^{K}p_{k_n}'\epsilon_{k_n}} = \sum_{A=1}^{K}\mathcal{O}(\text{Accepted})p_{A,\text{eff}}' \quad (15) \\
&= \sum_{A=1}^{K}\mathcal{O}(\text{Accepted})w_{\text{analytic}}^{\text{break}}p_{A,\text{eff}} = \mathbb{E}_A\left[w_{\text{analytic}}^{\text{break}}\mathcal{O}(\text{Accepted})\right],
\end{aligned}$$

where the step from the first to the second line is recognizing the infinite sum over rejections as the geometric series. Note that, in the last line, the expectation value is only to be taken over accepted hadrons. In appendix A, we include the full proof that the stochastic and analytic weights provide statistically identical results at the level of observables for the case of a fragmentation chain with multiple accepted string breaks; we furthermore discuss the logic behind the introduction of stochastic weights in some additional pedagogical detail.

We highlight that the main advantage of stochastic weights resides in avoiding explicit cumbersome calculations of effective probabilities. Conversely, the main drawback is that the weights will have a higher variance, and thus the reweighted samples will have lower effective statistics. This is the price to pay when trading explicit integration for a Monte Carlo estimation of the filter efficiencies.

For an event composed of a series of rejected and accepted fragmentations, one can compute the full analytic weight as the product of the individual weights for the accepted fragmentations, eq. (12), and the stochastic weight as the product of individual weights for the accepted *and rejected* fragmentations, eq. (9). These can be grouped in terms of the string break species and, for the stochastic weights, there is also no need to distinguish between accepted and rejected breaks. All the weights thus take the general form

$$w = \prod_{k=1}^{K}w_k^{n_k}, \quad (16)$$

where for the analytic prescription $n_k$ is the number of accepted fragmentations for species $k$, and for the stochastic prescription it is the number of accepted *and rejected* fragmentations. In practice, however, the weight is re-arranged in terms of blocks for which we can make use of the different conditional probabilities already specified in PYTHIA 8, as we discuss in the next section.

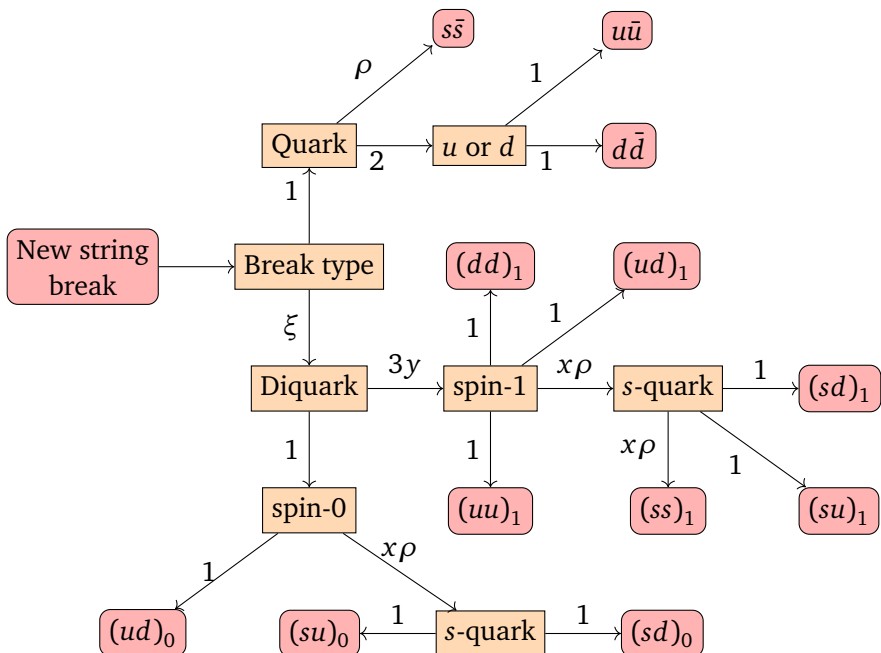

Figure 4: Sketch of the full flavor selection part of the string breaking algorithm, for the simple model for diquark production introduced in section 2. Note that the actual algorithm implemented in PYTHIA 8 differs from the one sketched here (see main text for details).

## 2.4 String break weight including diquarks

A more useful alternative to eq. (16) is the generalized version of eq. (5), where we consider the total weight for each possible binary branching in the string break algorithm. The general structure of the weight is thus

$$w = \prod_i w_i = \prod_i \left(\frac{p'_i}{p_i}\right)^{n_i} \left(\frac{1-p'_i}{1-p_i}\right)^{N_i-n_i}. \tag{17}$$

Unlike the discussion in previous sections, the product in the above equation is over branchings in the algorithm, not over the individual species nor over string breaks, and $p_i$ is the probability of "success" given a draw of branching $i$ (a generalization of $p_{s\bar{s}|s}$ considered in the previous sections). Although one could consider either the stochastic or the analytic probabilities, it is more straightforward to avoid the explicit introduction of effective probabilities and instead focus on the stochastic prescription, where this block structure is particularly useful. A sketch of the algorithm flow is shown in fig. 4, which will be explained in the following.

Note that the quark-only tree in fig. 2, on which the discussion in section 2.1 was based, is now the small branch at the top of fig. 4. Prior to this quark-only branch the decision on whether to split into a quark–anti-quark pair $q\bar{q}$ or a diquark–anti-diquark[9] pair $\mathcal{Q}\bar{\mathcal{Q}}$ must be taken. This prior decision is governed by the overall suppression factor $\xi$. In the diquark branch, the produced diquarks can be either in a spin-1 or spin-0 combination, where the former is suppressed by a factor $3y$, where the factor 3 comes from counting the spin states. Finally, both spin-1 and spin-0 diquarks can contain an $s$-quark, suppressed by a factor $x\rho$. The diquark $(ss)_1$ is thus suppressed by $(x\rho)^2$.

---

[9]We adopt the following notation for diquarks: a general diquark is denoted $\mathcal{Q}$, and an anti-diquark $\bar{\mathcal{Q}}$. If both flavors are specified, we denote it $(qq')$, or $(\bar{q}\bar{q}')$. Attributes are denoted by subscript. So, for example, an up-down diquark with spin-0 becomes $(ud)_0$ and a diquark-antidiquark pair conditioned on one quark in each being $s$, and the combination being spin-0, is denoted $\mathcal{Q}\bar{\mathcal{Q}}_{s0}$.

This sketch of the algorithm is different from what is implemented in PYTHIA 8, which collects all diquark production decisions into one large list with the fully combined weights rather than relying on successive either/or decisions. Representing the algorithm in terms of successive decisions, however, makes the equation for the weight calculation more legible, while the outputs of the two algorithms remain equivalent.

Changing the notation slightly from section 2.1, we will denote with $n_i$ the number of breakings of type $i$ (*e.g.*, $n_{s\bar{s}}$ for number of $s\bar{s}$ breakings, and $n_{\mathcal{Q}\bar{\mathcal{Q}}_1}$ for the number of spin-1 diquark breakings), with the special case $n$ denoting the total number of string breaks. Since we are considering the stochastic weights, the number of breakings of a given type now includes both accepted and rejected fragmentations. We will also provide indices to probabilities in the same vein. The master equation (*i.e.*, the expanded version of eq. (17)) will now consist of terms of the type:

$$w[x, y] \equiv \left(\frac{p'_{x|y}}{p_{x|y}}\right)^{n_x} \left(\frac{1 - p'_{x|y}}{1 - p_{x|y}}\right)^{n_y - n_x}, \tag{18}$$

where $x$ and $y$ are the flavors resulting from the string break and the conditional respectively. The full equation is then:

$$w_{\text{stochastic}} = \underbrace{w[\mathcal{Q}\bar{\mathcal{Q}}, \text{all}]}_{\mathcal{Q}\bar{\mathcal{Q}} \text{ from all breaks}} \times \underbrace{w[s\bar{s}, q\bar{q}]}_{s\bar{s} \text{ from } q\bar{q} \text{ breaks}} \times \underbrace{w[\mathcal{Q}\bar{\mathcal{Q}}_1, \mathcal{Q}\bar{\mathcal{Q}}]}_{\mathcal{Q}\bar{\mathcal{Q}}_1 \text{ from all } \mathcal{Q}\bar{\mathcal{Q}} \text{ breaks}} \tag{19}$$
$$\times \underbrace{w[\mathcal{Q}\bar{\mathcal{Q}}_{s0}, \mathcal{Q}\bar{\mathcal{Q}}_0]}_{\mathcal{Q}\bar{\mathcal{Q}}_{s0} \text{ from all } \mathcal{Q}\bar{\mathcal{Q}} \text{ breaks}} \times \underbrace{w[\mathcal{Q}\bar{\mathcal{Q}}_{s1}, \mathcal{Q}\bar{\mathcal{Q}}_1]}_{\mathcal{Q}\bar{\mathcal{Q}}_{s1} \text{ from all } \mathcal{Q}\bar{\mathcal{Q}}_1 \text{ breaks}} \times \underbrace{w[(ss)_1, \mathcal{Q}\bar{\mathcal{Q}}_{s1}]}_{(ss)_1 \text{ from all } \mathcal{Q}\bar{\mathcal{Q}}_{s1} \text{ breaks}}.$$

The probabilities $p_{x|y}$ associated with each term can be written in terms of the parameters $\rho, \xi, x, y$ as

$$p_{\mathcal{Q}\bar{\mathcal{Q}}|\text{all}} = \frac{\xi}{1 + \xi} \qquad (\mathcal{Q}\bar{\mathcal{Q}} \text{ break from any string break}), \tag{20a}$$

$$p_{s\bar{s}|q\bar{q}} = \frac{\rho}{2 + \rho} \qquad (s\bar{s} \text{ break from any } q\bar{q} \text{ break}), \tag{20b}$$

$$p_{\mathcal{Q}\bar{\mathcal{Q}}_1|\mathcal{Q}\bar{\mathcal{Q}}} = \frac{3y}{1 + 3y} \qquad (\mathcal{Q}\bar{\mathcal{Q}}_1 \text{ break from any } \mathcal{Q}\bar{\mathcal{Q}} \text{ break}), \tag{20c}$$

$$p_{\mathcal{Q}\bar{\mathcal{Q}}_{s0}|\mathcal{Q}\bar{\mathcal{Q}}_0} = \frac{2x\rho}{1 + 2x\rho} \qquad (\mathcal{Q}\bar{\mathcal{Q}}_{s0} \text{ break from any } \mathcal{Q}\bar{\mathcal{Q}}_0 \text{ break}), \tag{20d}$$

$$p_{\mathcal{Q}\bar{\mathcal{Q}}_{s1}|\mathcal{Q}\bar{\mathcal{Q}}_1} = \frac{x\rho}{3 + x\rho} \qquad (\mathcal{Q}\bar{\mathcal{Q}}_{s1} \text{ break from any } \mathcal{Q}\bar{\mathcal{Q}}_1 \text{ break}), \tag{20e}$$

$$P_{(ss)_1|\mathcal{Q}\bar{\mathcal{Q}}_{s1}} = \frac{x\rho}{2 + x\rho} \qquad ((ss)_1 \text{ break from any } \mathcal{Q}\bar{\mathcal{Q}}_{s1} \text{ break}). \tag{20f}$$

One can see in eq. (19) that all the individual blocks of weights are written in terms of one of the two outcomes of the branching (*e.g.*, $\mathcal{Q}\bar{\mathcal{Q}}_{s1}$), and the total number of draws for that branching (*e.g.*, all $\mathcal{Q}\bar{\mathcal{Q}}_1$ breaks). All the probabilities except $p_{\mathcal{Q}\bar{\mathcal{Q}}|\text{all}}$ are thus conditional probabilities. The modular decomposition of the total weight into conditional branching steps reflects the underlying tree structure of the algorithm and makes it straightforward to extend the approach to other hadronization models or even unrelated multi-step generative procedures.

With eqs. (19) and (20) in hand, one can now calculate the event weights at string break level between values of parameters $\rho, \xi, x,$ and $y$, as long as the quantity of each type of string break is retained on an event-by-event basis.

To obtain the analytic weights for this more general case, one simply needs to replace the probabilities by the corresponding effective probabilities. A word of caution, however, is to be careful regarding connections between blocks. For example, including the $\eta$ and $\eta'$ filters

re-defines $p_s \mapsto \frac{\rho_{\text{eff}}}{2+\rho_{\text{eff}}}$, as before, but also modifies $p_{Q\bar{Q}} \mapsto \frac{\xi_{\text{eff}}}{1+\xi_{\text{eff}}}$ with $\xi_{\text{eff}} = \xi \frac{(2+\rho)}{(2+\rho_{\text{eff}})} \geq \xi$ to account for the extra diquark breaks arising from $\eta$ and $\eta'$ suppression. Due to these complications, and because effective statistics is not an issue in the presented application, we do not write the analytic weights up in the following, but focus exclusively on the development of stochastic weights.

## 2.5 Filters for baryon production

If neither popcorn splitting nor junctions are considered, as is the case in this work, a baryon is always formed by the combination of a diquark and a quark from two adjacent string breaks. The diquark and quark combination needs to be projected to a spin × flavor symmetric SU(6) representation, with the baryon belonging either to the octet or the decuplet subsets of the 56-multiplet. For a given diquark and quark combination, the relative probability between octet and decuplet is fixed by the Clebsch–Gordan coefficients up to an additional parameter $\epsilon_{10/8}$ allowing for further suppression of the decuplet relative to the octet.[10] Like other filter probabilities, we consider this parameter fixed and not subject to reweighting.

A filter is applied to ensure that the diquark–quark samples combined into baryons respect the SU(6) Clebsch–Gordan coefficients both for the octet and the decuplet. In practice, this implies that some diquark–quark combinations are rejected at random by the filter to correct their frequency relative to other diquark–quark combinations. This frequency, and thus the filter, is determined by the sum of the Clebsch–Gordan coefficient of that combination to the octet and the decuplet. The physics of the spin × flavor projection is explained in the following.

A baryon state is symmetric in spin × flavor. Therefore, to produce a baryon from a quark and a diquark, the probability for them to join in such a state must be taken into account. Take, for example, a $(uu)_1$-diquark (subscript indicating spin) combining with another $u$-quark. Since all quarks are identical, an octet state cannot be reached, and therefore trivially has a weight of 0. The decuplet has two baryons with spin × flavor wave functions:

$$
\begin{aligned}
|\Delta^{++}_{\frac{3}{2}\frac{3}{2}}\rangle &= (u\uparrow u\uparrow u\uparrow), \\
|\Delta^{++}_{\frac{3}{2}\frac{1}{2}}\rangle &= \frac{1}{\sqrt{3}}(u\uparrow u\uparrow u\downarrow + u\uparrow u\downarrow u\uparrow + u\downarrow u\uparrow u\uparrow),
\end{aligned}
\tag{21}
$$

where the subscript indicates the usual spin states $|jm\rangle$. We write the diquark wave function as:

$$
|(uu)_1\rangle = (u\uparrow u\uparrow).
\tag{22}
$$

The total weight for the decuplet is the sum

$$
w_{(uu)_1+u,10} = 1^2 + \left(\frac{1}{\sqrt{3}}\right)^2 = \frac{4}{3},
\tag{23}
$$

identical to what a normal Clebsch–Gordan table for combining spins $1 \times 1/2$ (identical particles) would give. Since one would like to avoid weights larger than unity, and an accept-reject procedure only requires relative weights, we choose an overall normalization of weights making this weight unity. Following this procedure for all combinations produces the table of weights shown in table 1, reproduced from ref. [29]. A full derivation of all weights is given in appendix B.

The Clebsch–Gordan filter can now be applied for each individual diquark–quark combination. The PYTHIA 8 implementation of this filter is somewhat more involved, split in the two cases of either starting with a diquark and accepting/rejecting a quark, and starting with

---

[10]In PYTHIA 8, the parameter $\epsilon_{10/8}$ has the name `StringFlav:decupletSup` and a default value of 1, *i.e.*, no further suppression of the decuplet.

Table 1: Table of SU(6) spin × flavor Clebsch–Gordan weights for joining a quark and a diquark into a baryon in the octet or decuplet respectively. Table reproduced from ref. [29]; a full derivation of all weights is given in appendix B.

| Diquark | $(ud)_0$ | | $(ud)_1$ | | $(uu)_1$ | |
| --- | --- | --- | --- | --- | --- | --- |
| Quark | $u$ or $d$ | $s$ | $u$ or $d$ | $s$ | $u$ | $d$ or $s$ |
| Octet | $\frac{3}{4}$ | $\frac{1}{2}$ | $\frac{1}{12}$ | $\frac{1}{6}$ | 0 | $\frac{1}{6}$ |
| Decuplet | 0 | 0 | $\frac{2}{3}$ | $\frac{1}{3}$ | 1 | $\frac{1}{3}$ |

a quark and accepting/rejecting a diquark. This allows for a renormalization of coefficients, which in turn makes the implementation more efficient. However, the produced result is equivalent to the simplified algorithm as described above.

Additional $\Lambda$–$\Sigma^0$ mixing is explicitly taken into account, much like $\eta$–$\eta'$ mixing but with non-tuneable parameters. In general, if no parameters of baryon production from a quark and diquark are reweighted, and because the filters can be incorporated implicitly, the stochastic weights defined in eq. (19) need not be modified to account for baryon production. Analogous analytic weights may be defined at the expense of cumbersome calculations to incorporate the Clebsch–Gordan coefficient filtering but allowing for reduced loss in effective statistics.

## 2.6 Kinematic filtering: The effect of `finalTwo`

In the previous sections, we have detailed how to reweight between different flavor parameter choices. Focusing on the stochastic prescription, we can use eq. (19) to reweight any observable over produced mesons and baryons by taking into account all sampled breaks, including not only the observed mesons and baryons, but also the samples that were rejected by different filters, *e.g.*, those that account for $\eta$ and $\eta'$ suppression and Clebsch–Gordan rejection.

There is, however, an additional filter that needs to be taken into account: the `finalTwo` filter that ensures that all the hadrons can be produced on-shell and that the overall fragmentations follow the left-right symmetric Lund fragmentation function.[11] If a fragmentation chain composed of all sampled hadrons (accepted by the other filters) fails to pass the `finalTwo` condition, the entire generated fragmentation chain is rejected, and the simulation of hadronization starts anew.

To appropriately account for the `finalTwo` filter efficiency, one can follow the complete stochastic prescription and store the weights of all sampled fragmentations, both accepted and rejected by `finalTwo`, as was done for the kinematic weights in ref. [18]. It is also, in principle, possible to accommodate analytic weights if calculated, by treating `finalTwo` as a filter, with an efficiency calculated in the stochastic approach.

A completely analytic prescription is likely impossible for `finalTwo` filtering, however, since the filter efficiencies depend non-trivially on the full set of kinematics and flavor assignments of the fragmentation chain.

# 3 Validation

As with the kinematic reweighting method presented in ref. [18], the goal of the presented method is to produce desired distributions using alternative weights, rather than generating new samples for each set of alternative parameter values.

---

[11]This is the requirement that one should, on average, obtain the same result regardless of whether one chooses to hadronize a string from the left, or from the right.

An implementation of the algorithm, as explained in the preceding sections, has been implemented in the PYTHIA 8 Monte Carlo event generator.[12] The distribution includes examples for users; see the online documentation.

In this section, we validate the method by generating samples of $10^7$ events using PYTHIA 8 configured with a set of baseline parameter values and at the same time calculating per-event weights $w_i'$ that each correspond to an alternative set of parameter values. We then compare the $w_i'$-weighted distributions to those obtained by generating events with the alternative parameter values set as the baseline.

We do this for parameters $\rho, \xi, x$, and $y$. The top panels in figs. 5 to 8 demonstrate that the chosen observables, defined below, are sensitive to changes in $\rho, \xi, x$, and $y$, respectively. The bottom panels in figs. 5 to 8 show the agreement between the $w_i'$-weighted distributions and those generated with alternative parameter values set as the baseline. We also vary $\rho$ and $x$ and $\xi, \rho, x$, and $y$ simultaneously, and figs. 9 and 10 show the analogous plots for these cases. Whenever not explicitly stated, the parameters are set to the values used in PYTHIA prior to the Monash tune, $\xi = 0.09, \rho = 0.19, x = 1$, and $y = 0.027$ [30].[13]

## 3.1 Validation simulations

The $w'$-weighted distributions are generally in good agreement with the distributions obtained directly from the baseline parameter values. To be able to achieve this agreement the support of the modified string-break probability distribution needs to lie entirely within the support of the baseline distribution—that is, the baseline distribution must be nonzero wherever the modified distribution is nonzero, see also ref. [18]. Such lack of support is evident in the tail of the $s$-quark break distribution in fig. 9 (bottom right), where the method requires large weights and samples the phase space poorly, as illustrated in fig. 11.

In performing flavor parameter variations using the method of reweighting it is therefore important to select appropriate baseline values for the parameters. Whether the chosen values are appropriate can be evaluated with the test statistics $1-\mu$ and $n_{\rm eff}/N$, where $N$ is the number of events, $\mu$ is the average weight [18]

$$\mu \equiv \sum_{i=1}^{N} \frac{w_i'}{N}, \tag{24}$$

and $n_{\rm eff}$ the effective sample size,

$$n_{\rm eff} \equiv \frac{\left(\sum_{i=1}^{N} w_i'\right)^2}{\sum_{i=1}^{N} w_i'^2}. \tag{25}$$

Significant deviations of $1-\mu$ from zero or $n_{\rm eff}/N$ from unity may indicate insufficient coverage of the alternative parameter values by the baseline distribution [18]. Table 2 shows the $1-\mu$ and $n_{\rm eff}/N$ values corresponding to the various figures.

It is important to note that it can be quite difficult to simulate sufficient statistics to compensate for a lack of coverage in the probability distribution, especially in edge cases. To illustrate, we have repeated the tests with one tenth the sample size ($10^6$ events) using the same random number seed and show the resulting $s$-quark break distributions in fig. 12. In

---

[12]The implementation has been in place in the public version of PYTHIA 8 since version 8.312. The newest version can be downloaded freely from https://www.pythia.org/.

[13]The other relevant generation parameters used in each PYTHIA 8 sample are `Beams:idA = 11`, `Beams:idB = -11`, `Beams:eCM = 91.189`, `PDF:lepton = off`, `WeakSingleBoson:ffbar2gmZ = on`, `23:onMode = off`, `23:onIfAny = 1 2 3 4 5`, `HadronLevel:Decay = off`, and `StringFlav:popcornRate = 0`.

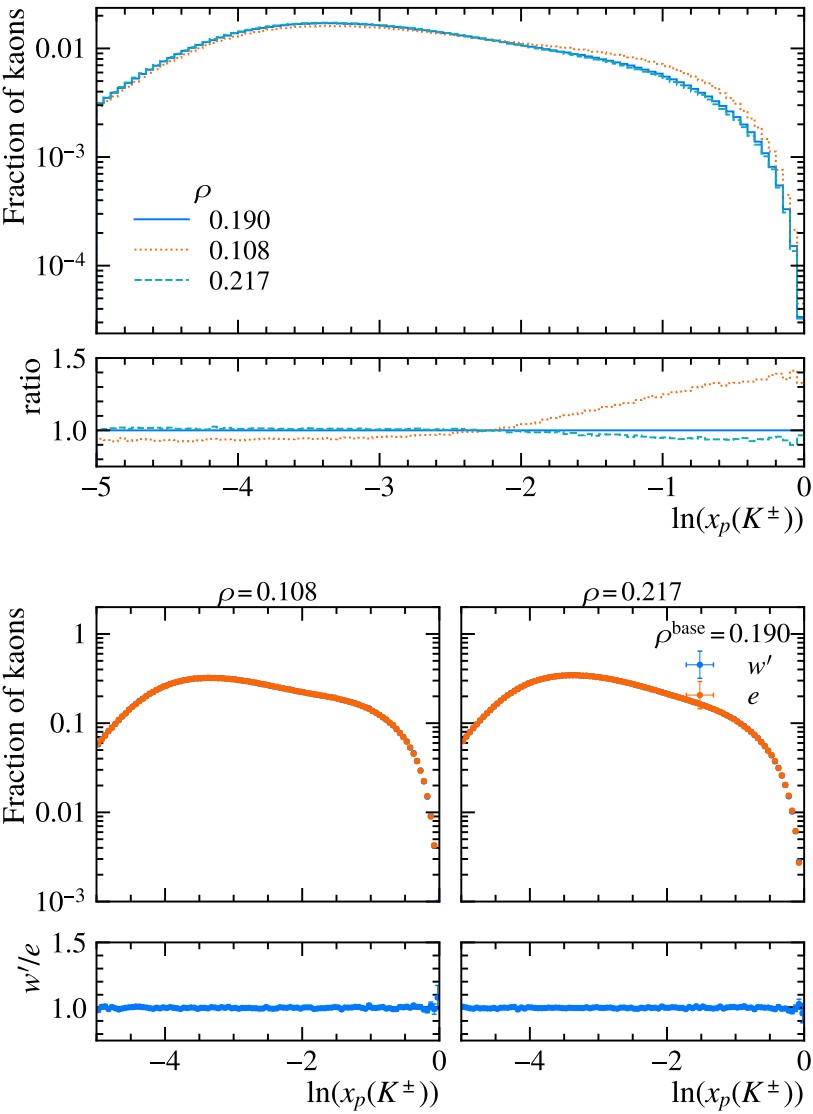

Figure 5: Comparison of $\ln(x_p(K^{\pm}))$ distributions, where $x_p$ is the beam momentum fraction (see eq. (4) of ref. [30]), shown as fractions of the total number of kaons in an event, when the parameter $\rho$ is (top) explicitly set to different values or (bottom) varied using different methods. In the top panel, the lower row shows the ratios of the distributions generated with various values of $\rho$ to that generated with $\rho = 0.190$. In the bottom panel, the distributions labeled $e$ were generated with the value of the parameter $\rho$ explicitly set to (left) 0.108 and (right) 0.217. The distributions labeled $w'$ are all taken from the same sample generated with $\rho = \rho^{\text{base}} = 0.190$ but with different sets of alternative event weights $w'$ corresponding to the alternative values of $\rho$. The bottom row shows the ratios of the latter distributions to the former. Statistical error bars shown.

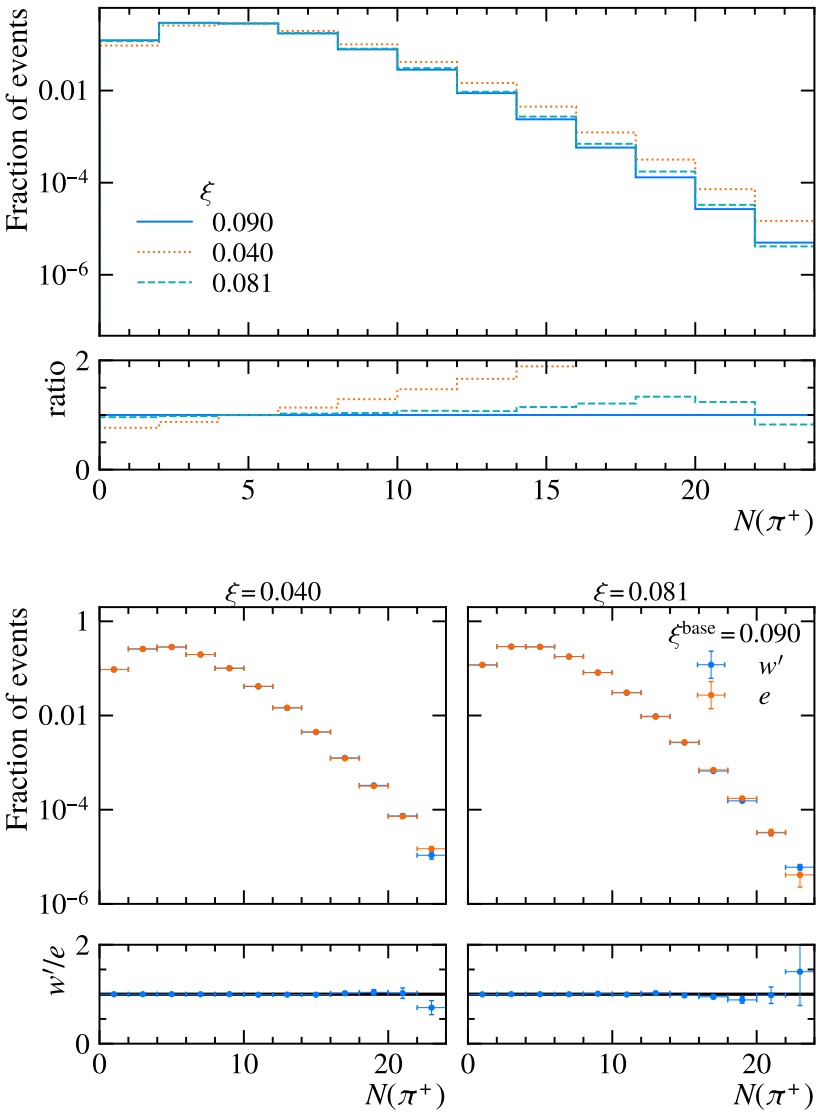

Figure 6: Comparison of the distributions of the number of pions (charge-conjugation implied) in an event, shown as fractions of the total number of events in a sample, when the parameter $\xi$ is explicitly set to (top) different values or (bottom) varied using different methods. In the top panel, the lower row shows the ratios of the distributions generated with various values of $\xi$ to that generated with $\xi = 0.090$. In the bottom panel, the distributions labeled $e$ were generated with the value of the parameter $\xi$ explicitly set to (left) 0.040 and (right) 0.081. The distributions labeled $w'$ are all taken from the same sample generated with $\xi = \xi^{\text{base}} = 0.090$, but with different sets of alternative event weights $w'$ corresponding to the alternative values of $\xi$. The bottom row shows the ratios of the latter distributions to the former. Statistical error bars shown.

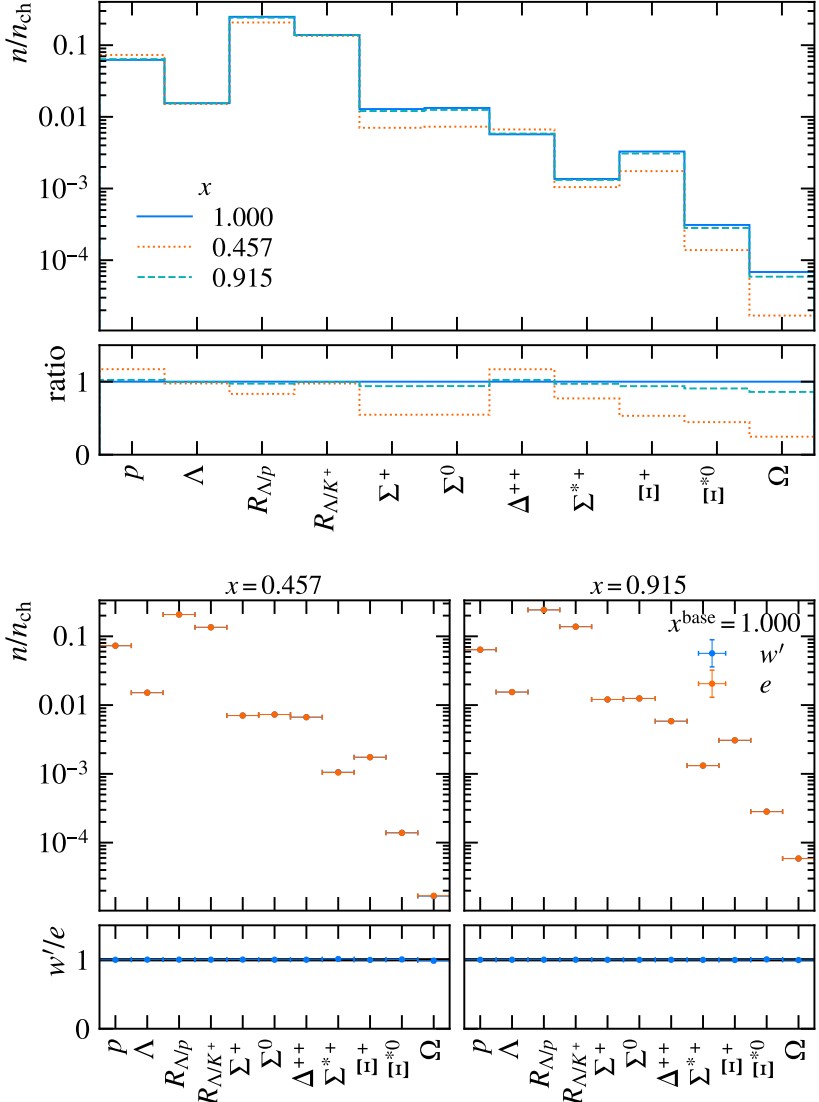

Figure 7: Comparison of the relative rates of various particle species when the parameter $x$ is (top) explicitly set to different values or (bottom) varied using different methods. The ratios are with respect to all charged particles in the event unless otherwise indicated in the bin label, following fig. 5 of ref. [30]. In the top panel, the lower row shows the ratios of the distributions generated with various values of $x$ to that generated with $x = 1.000$. In the bottom panel, the distributions labeled $e$ were generated with the value of the parameter $x$ explicitly set to (left) 0.457 and (right) 0.915. The distributions labeled $w'$ are all taken from the same sample generated with $x = x^{\text{base}} = 1.000$, but with different sets of alternative event weights $w'$ corresponding to the alternative values of $x$. The bottom row shows the ratios of the latter distributions to the former. Statistical error bars shown.

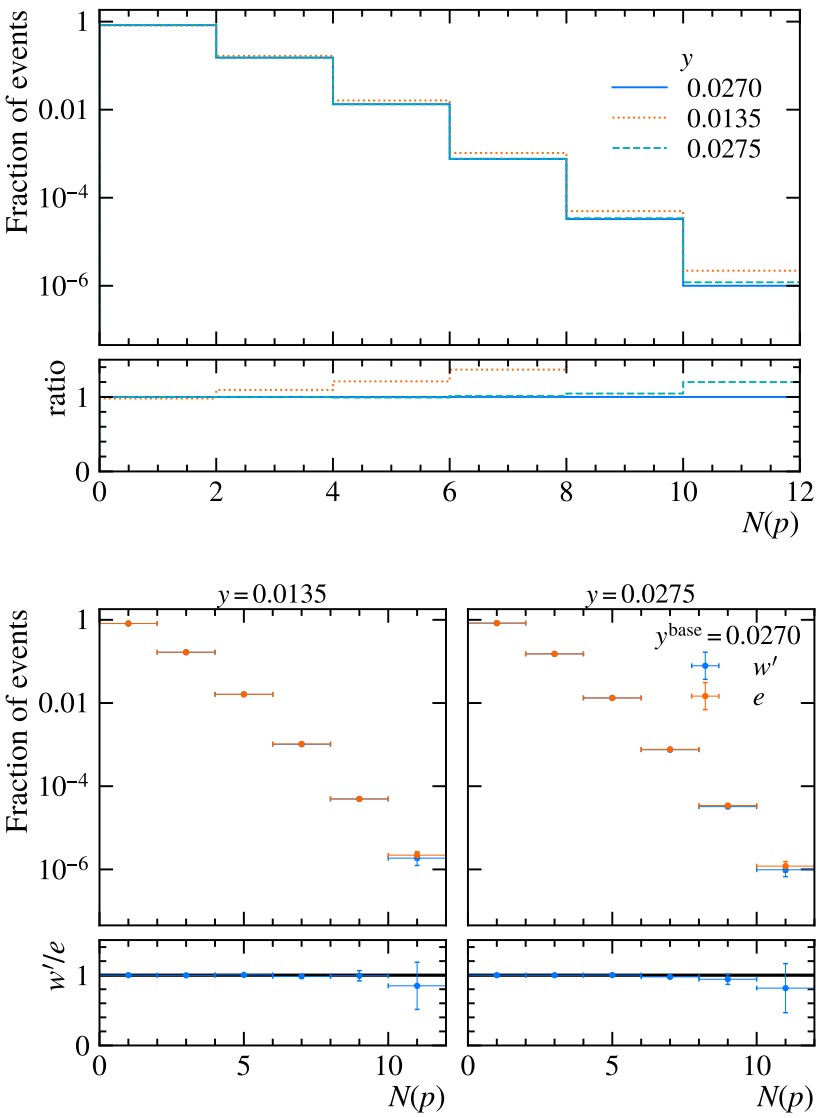

Figure 8: Comparison of the distributions of the number of protons (charge-conjugation implied) in an event, shown as fractions of the total number of events in a sample, when the parameter $y$ is (top) explicitly set to different values or (bottom) varied using different methods. In the top panel, the lower row shows the ratios of the distributions generated with various values of $y$ to that generated with $y = 0.0270$. In the bottom panel, the distributions labeled $e$ were generated with the value of the parameter $y$ explicitly set to (left) 0.0135 and (right) 0.0275. The distributions labeled $w'$ are all taken from the same sample generated with $y = y^{\mathrm{base}} = 0.0270$, but with different sets of alternative event weights $w'$ corresponding to the alternative values of $y$. The bottom row shows the ratios of the latter distributions to the former. Statistical error bars shown.

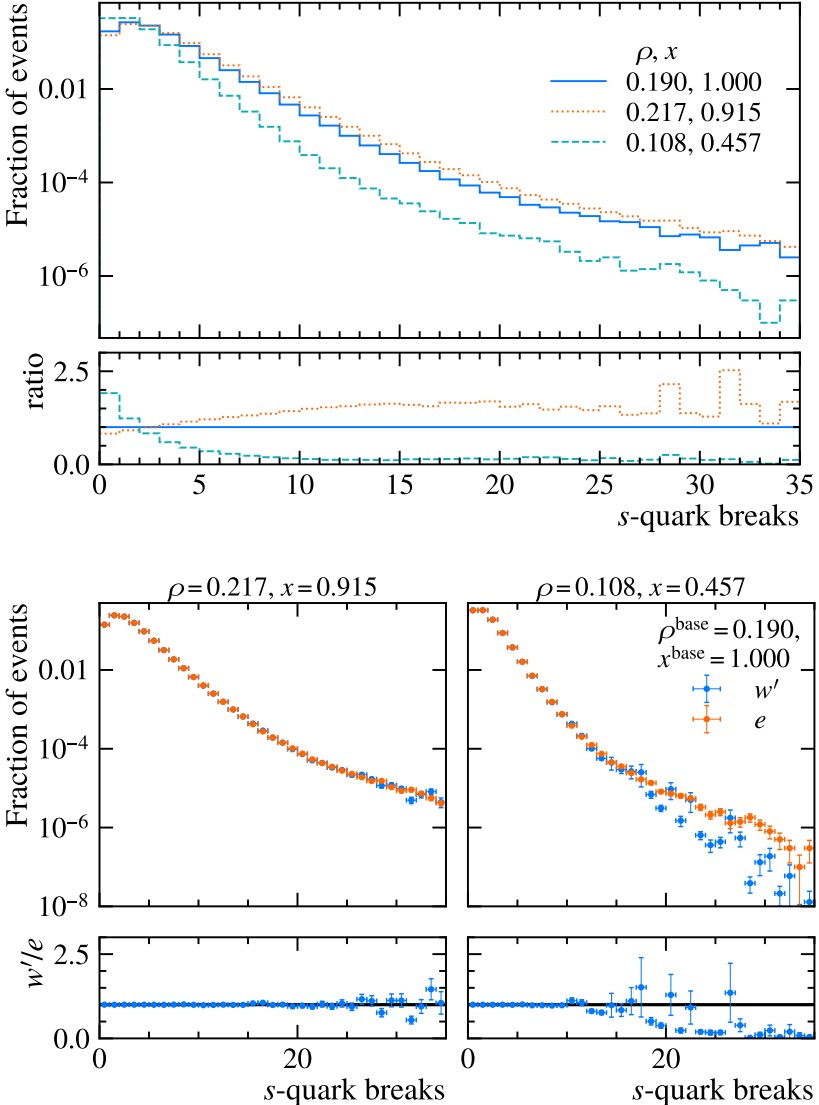

Figure 9: Comparison of the distributions of the number of $s$-quark breaks in an event, shown as fractions of the total number of events in a sample, when the parameters $\rho$ and $x$ are (top) explicitly set to different values or (bottom) simultaneously varied using different methods. In the top panel, the lower row shows the ratios of the distributions generated with various values of $\rho$ and $x$ to that generated with $\rho = 0.190$ and $x = 1.000$. In the bottom panel, the distributions labeled $e$ were generated with the values of $\rho, x$ explicitly set to (left) $0.217, 0.915$ and (right) $0.108, 0.457$. The distributions labeled $w'$ are all taken from the same sample generated with $\rho = \rho^{\text{base}} = 0.190$ and $x = x^{\text{base}} = 1.000$ but with different sets of alternative event weights $w'$ corresponding to the alternative values of $\rho$ and $x$. The bottom row shows the ratios of the latter distributions to the former. Statistical error bars shown.

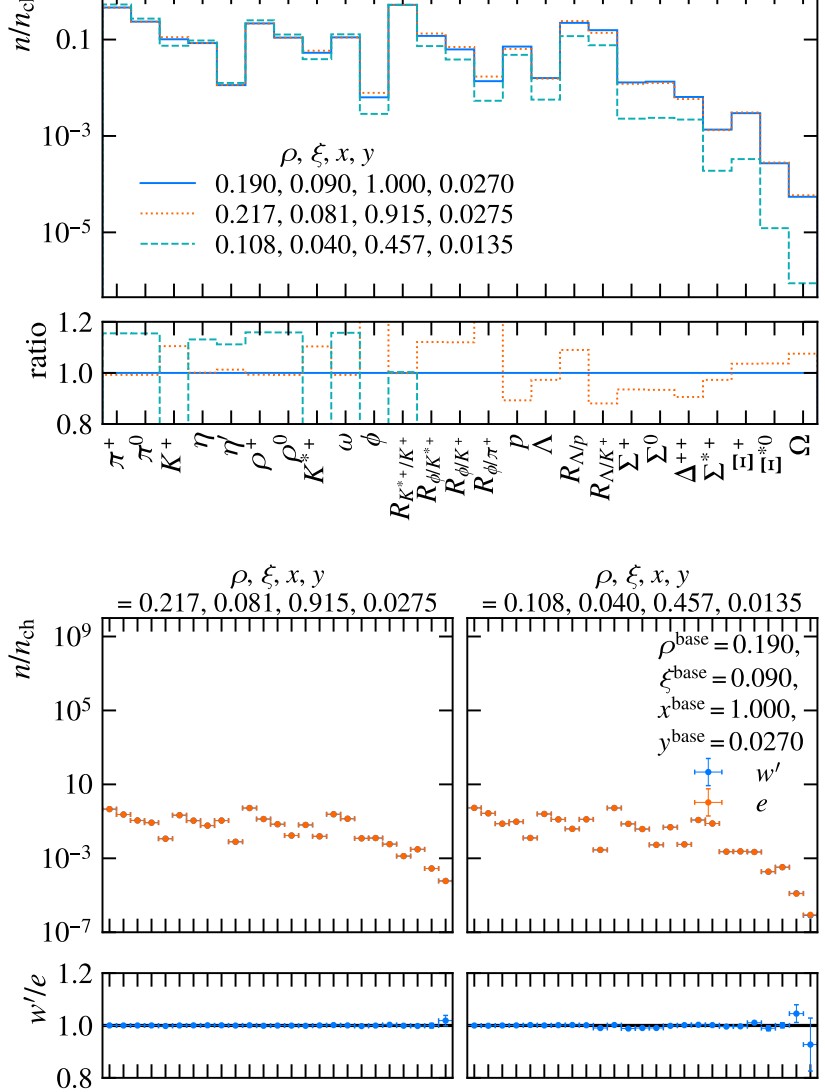

Figure 10: Comparison of the relative rates of various particle species when the parameters $\rho, \xi, x$, and $y$ are (top) explicitly set to various values or (bottom) simultaneously varied using different methods. The ratios are with respect to all charged particles in the event unless otherwise indicated in the bin label, following fig. 5 of ref. [30]. In the top panel, the lower row shows the ratios of the distributions generated with various values of $\rho, \xi, x$, and $y$ to that generated with $\rho = 0.190, \xi = 0.090, x = 1.000$, and $y = 0.0270$. In the bottom panel, the distributions labeled $e$ were generated with the values of the parameters $\rho, \xi, x$, and $y$ explicitly set to (left) $\rho, \xi, x, y = 0.217, 0.081, 0.915, 0.0275$ and (right) $\rho, \xi, x, y = 0.108, 0.040, 0.457, 0.0135$. The distributions labeled $w'$ are all taken from the same sample generated with $\rho = \rho^{\text{base}} = 0.190, \xi = \xi^{\text{base}} = 0.090, x = x^{\text{base}} = 1.000$, and $y = y^{\text{base}} = 0.0270$ but with different sets of alternative event weights $w'$ corresponding to the alternative values of $\rho, \xi, x$, and $y$. The bottom row shows the ratios of the latter distributions to the former. Statistical error bars shown.

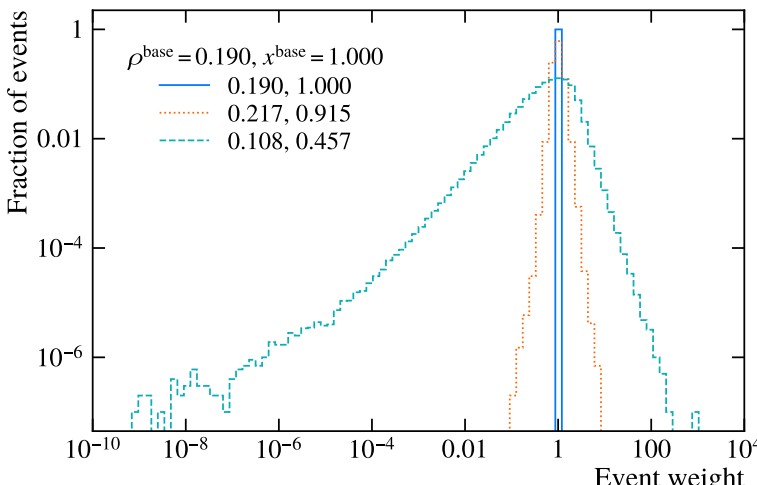

Figure 11: Distributions of the event weights $w'$ for reweighting from $\rho = \rho^{\text{base}} = 0.190$ and $x = x^{\text{base}} = 1.000$, shown on a log-log scale with varying bin sizes. The weights when $\rho = \rho^{\text{base}} = 0.190$ and $x = x^{\text{base}} = 1.000$ are exactly equal to 1.

Table 2: Difference of mean weight $\mu$ from one and the ratio of the effective sample size $n_{\text{eff}}$ to the number of generated events $N = 10^7$ for the listed variations, shown with statistical errors; see the text. The values $1 - \mu = 0$ and $n_{\text{eff}}/N = 1$ correspond to the baseline case where all weights are unity.

| variation | $1 - \mu$ | $n_{\text{eff}}/N$ | figure |
|---|---|---|---|
| $\rho^{\text{base}} = 0.190$ | 0 | 1 | |
| $\rho = 0.108$ | $-(3.5 \pm 2.7) \times 10^{-4}$ | $7.4 \times 10^{-1}$ | fig. 5 |
| $\rho = 0.217$ | $(1.5 \pm 0.7) \times 10^{-4}$ | $9.8 \times 10^{-1}$ | |
| $\xi^{\text{base}} = 0.090$ | 0 | 1 | |
| $\xi = 0.040$ | $(3.9 \pm 3.4) \times 10^{-4}$ | $6.7 \times 10^{-1}$ | fig. 6 |
| $\xi = 0.081$ | $(2.0 \pm 4.5) \times 10^{-5}$ | $9.9 \times 10^{-1}$ | |
| $x^{\text{base}} = 1.000$ | 0 | 1 | |
| $x = 0.457$ | $(2.0 \pm 1.3) \times 10^{-4}$ | $8.9 \times 10^{-1}$ | fig. 7 |
| $x = 0.915$ | $(2.2 \pm 1.7) \times 10^{-5}$ | $1.0 \times 10^{0}$ | |
| $y^{\text{base}} = 0.0270$ | 0 | 1 | |
| $y = 0.0135$ | $(1.8 \pm 1.0) \times 10^{-4}$ | $9.5 \times 10^{-1}$ | fig. 8 |
| $y = 0.0275$ | $-(5.1 \pm 3.2) \times 10^{-6}$ | $1.0 \times 10^{0}$ | |
| $\rho^{\text{base}} = 0.190, x^{\text{base}} = 1.000$ | 0 | 1 | |
| $\rho = 0.108, x = 0.457$ | $(5.3 \pm 3.7) \times 10^{-4}$ | $6.1 \times 10^{-1}$ | figs. 9 and 11 |
| $\rho = 0.217, x = 0.915$ | $(4.2 \pm 6.4) \times 10^{-5}$ | $9.8 \times 10^{-1}$ | |
| $\rho^{\text{base}} = 0.190, \xi^{\text{base}} = 0.090,$ $x^{\text{base}} = 1.000, y^{\text{base}} = 0.0270$ | 0 | 1 | |
| $\rho = 0.108, \xi = 0.040,$ $x = 0.457, y = 0.0135$ | $-(4.9 \pm 8.1) \times 10^{-4}$ | $3.3 \times 10^{-1}$ | fig. 10 |
| $\rho = 0.217, \xi = 0.081,$ $x = 0.915, y = 0.0275$ | $(1.2 \pm 7.9) \times 10^{-5}$ | $9.7 \times 10^{-1}$ | |

Table 3: Difference of mean weight $\mu$ from one and the ratio of the effective sample size $n_{\text{eff}}$ to the number of generated events $N = 10^6$ for the listed variations, shown with statistical errors; see the text. The values $1 - \mu = 0$ and $n_{\text{eff}}/N = 1$ correspond to the baseline case where all weights are unity.

| variation | $1 - \mu$ | $n_{\text{eff}}/N$ | figure |
|---|---|---|---|
| $\rho^{\text{base}} = 0.190$ | 0 | 1 | |
| $\rho = 0.108$ | $-(1.4 \pm 0.9) \times 10^{-3}$ | $7.5 \times 10^{-1}$ | |
| $\rho = 0.217$ | $(5.5 \pm 2.2) \times 10^{-4}$ | $9.8 \times 10^{-1}$ | |
| $\xi^{\text{base}} = 0.090$ | 0 | 1 | |
| $\xi = 0.040$ | $(0.5 \pm 1.1) \times 10^{-3}$ | $6.7 \times 10^{-1}$ | |
| $\xi = 0.081$ | $(1.2 \pm 1.4) \times 10^{-4}$ | $9.9 \times 10^{-1}$ | |
| $x^{\text{base}} = 1.000$ | 0 | 1 | |
| $x = 0.457$ | $-(0.2 \pm 4.1) \times 10^{-4}$ | $8.9 \times 10^{-1}$ | |
| $x = 0.915$ | $-(1.9 \pm 5.4) \times 10^{-5}$ | $1.0 \times 10^0$ | |
| $y^{\text{base}} = 0.0270$ | 0 | 1 | |
| $y = 0.0135$ | $(1.7 \pm 3.1) \times 10^{-4}$ | $9.5 \times 10^{-1}$ | |
| $y = 0.0275$ | $-(0.6 \pm 1.0) \times 10^{-5}$ | $1.0 \times 10^0$ | |
| $\rho^{\text{base}} = 0.190, x^{\text{base}} = 1.000$ | 0 | 1 | |
| $\rho = 0.108, x = 0.457$ | $(0.8 \pm 1.1) \times 10^{-3}$ | $6.3 \times 10^{-1}$ | fig. 12 |
| $\rho = 0.217, x = 0.915$ | $(2.8 \pm 2.0) \times 10^{-4}$ | $9.8 \times 10^{-1}$ | |
| $\rho^{\text{base}} = 0.190, \xi^{\text{base}} = 0.090,$ $x^{\text{base}} = 1.000, y^{\text{base}} = 0.0270$ | 0 | 1 | |
| $\rho = 0.108, \xi = 0.040,$ $x = 0.457, y = 0.0135$ | $(1.8 \pm 2.2) \times 10^{-3}$ | $3.8 \times 10^{-1}$ | |
| $\rho = 0.217, \xi = 0.081,$ $x = 0.915, y = 0.0275$ | $(2.0 \pm 2.5) \times 10^{-4}$ | $9.7 \times 10^{-1}$ | |

this subsample, the ratio of reweighted to target tail distributions, for the case where the target parameter values differ greatly from the baseline, tends more dramatically toward zero than in the larger statistics sample. Put another way, generating a sample ten times the size of that used in fig. 12 still provides insufficient statistics to adequately capture the behavior of the tails, as can be seen by comparing the bottom right plots in figs. 9 and 12, though increasing the sample size does still *improve* the agreement. Note that the value of $1 - \mu$ for the subsample is $(0.8 \pm 1.1) \times 10^{-3}$, compared with $(5.3 \pm 3.7) \times 10^{-4}$ in table 2, quite small in both cases, indicative of the fact that the reweighting captures the target distribution quite well for the overwhelming bulk of events. In order to capture the behavior of distributional tails, it is likely far more computationally expedient to use baseline parameter values closer to those of the target reweighting, as contemplated in ref. [18], and the deviation of $n_{\text{eff}}/N$ from one should be taken seriously as an indicator of when to adjust the baseline parameter values in this way. The values of $1 - \mu$ and $n_{\text{eff}}/N$ for each subsample are shown in table 3 for comparison with table 2.

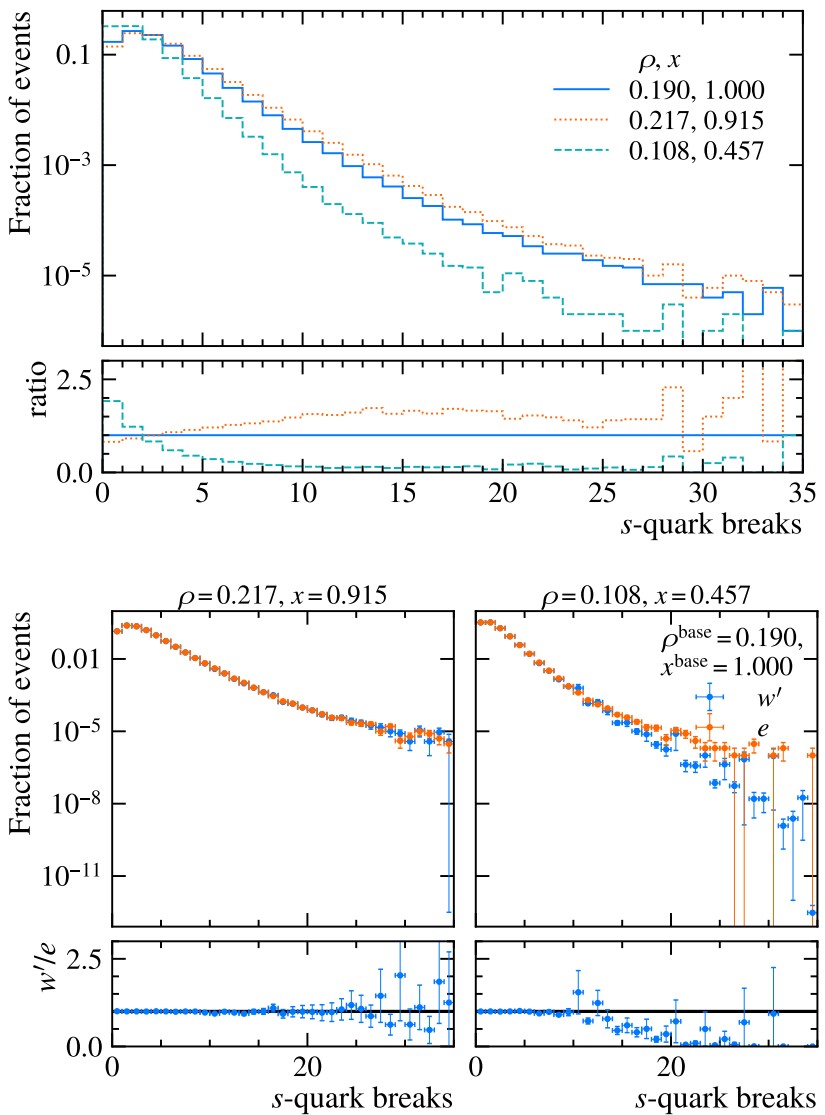

Figure 12: Comparison of the distributions of the number of $s$-quark breaks in an event, shown as fractions of the total number of events in a sample, when the parameters $\rho$ and $x$ are (top) explicitly set to different values or (bottom) simultaneously varied using different methods. This figure is identical to fig. 9 but drawn from a sample of only $10^6$ events, rather than $10^7$; refer to its caption for details.

## 3.2 Timing

The method of reweighting is universally faster than generating new samples with the alternative parameter values set explicitly. Furthermore, the time gains presented here are also greater than those presented for the method developed in ref. [18]. This is because the method introduced in this work does not depend on oversampling a target distribution using an accept/reject algorithm. When comparing the two methods one should note that there is an additional technical detail which differentiates between them. In ref. [18] we implemented an *in situ* method where the parameter variations must be defined before running the hadronization and a weight per parameter group is calculated event-by-event. This method is memory efficient but requires parameter variations to be specified prior to generation. However, it is also possible to implement a *post hoc* method, where sufficient information is stored per event

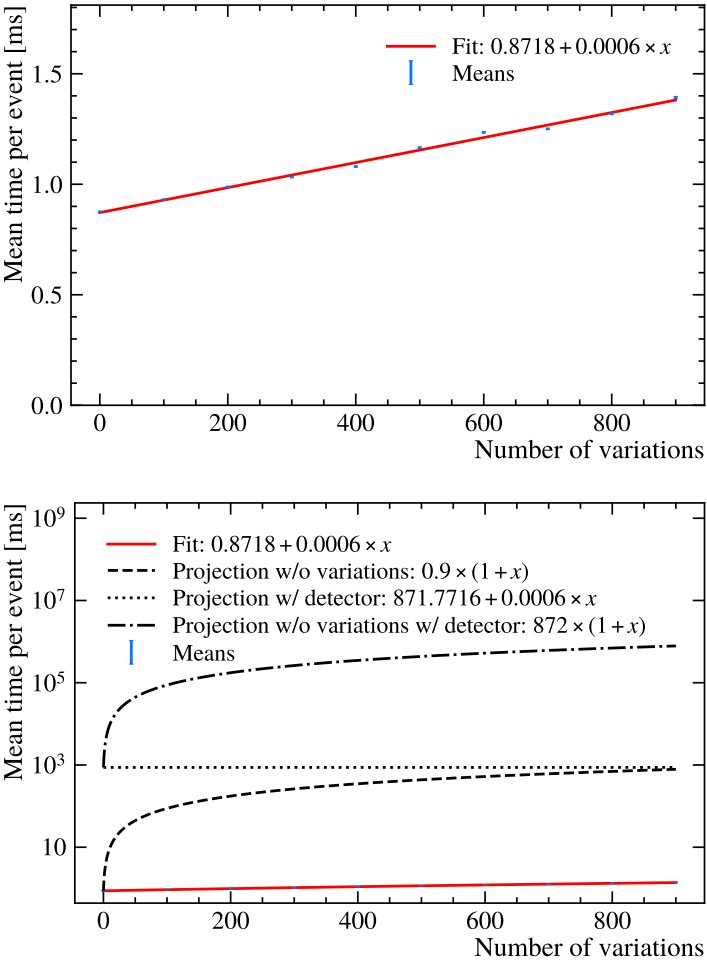

Figure 13: (Top) average time required to generate a single event as a function of the number of alternative parameter value sets for parameter $\rho$ and $x$ calculated during the generation. The error on each point is the standard error of the mean. The amount of time required to generate a single event increases linearly; the best-fit curve is shown in red, and its equation is given in the legend. (Bottom) the same, now including projections.

to calculate an event weight for any given parameter variations after the event is generated. Such a *post hoc* method was implemented in a different context in ref. [32]. For flavor parameter variations, the information needed per event to calculate *post hoc* weights is on the order of 20 integers, which has a sufficiently small memory footprint that the advantages of the *post hoc* method here far outweigh those of the *in situ* method. Consequently, we implement the flavor parameter reweighting of this work in PYTHIA 8 using the *post hoc* method.

To demonstrate the timing benefits of reweighting, we generate a set of 10 samples with $10^4$ events each, using the same PYTHIA 8 settings described above, where we calculate weights for additional alternative parameter values in each sample as each event is generated. We measure the time it takes to generate each event using a single 2.2 GHz Intel Xeon CPU. Figure 13 (top) shows the arithmetic mean of the time spent to generate a single event as a function of the number of alternative sets of values calculated for parameters $\rho$ and $x$. As shown, the marginal cost per additional parameter variation is $\approx 0.0006$ ms, and it takes $\approx 1.4$ ms to generate an event with 1,000 alternative values. Since it takes $\approx 0.9$ ms to generate an event with no alternative values, it would take $\approx 900$ ms to generate 1,000 separate events

with the alternative values set explicitly. These savings vary, depending on the parameters in question, but in all cases, they increase dramatically when one considers detector simulations, which often take $\approx 1,000$ times longer than the event generation; fig. 13 (bottom) shows a projection.

We emphasize that the timing considerations offered here are more universal than a statement about making Monte Carlo event generation faster. The reweighting procedure allows a single generated event sample to be reused across multiple parameter evaluations. This statement holds true also when the hadronization model is part of a larger simulation chain starting with a demanding matrix element generation and ending with an even more demanding detector simulation. A single event can be reweighted at any future point as long as the necessary information is retained, greatly increasing the statistical yield per CPU hour spent.

# 4 Conclusion and perspectives

In Monte Carlo event generator simulations of high-energy collisions, hadronization—and its associated theoretical uncertainties—is ubiquitous. While most other parts of the simulation tool chain today allow for efficient uncertainty estimation, often using various reweighting techniques, hadronization models have long been one of the last hold-outs relying on single-parameter predictions without variations. To our knowledge, our recent paper [18] on reweighting the kinematics selection in the Lund string model was the first in this space. In that sense, the present paper is a natural follow-up, dealing not only with a new set of model parameters, but also introducing new reweighting strategies which are more generally applicable than just for this particular model.

We have introduced a novel procedure to calculate and apply event weights that allow reweighting from one set of hadronization parameters for flavor selection to another, and have fully validated the procedure for simple fragmenting string systems, *i.e.*, without junctions or baryon production from the popcorn mechanism. The procedure is *post hoc* meaning it can be applied to a set of events already generated, without requiring the target parameters to be specified in advance. Instead, one needs to store only an additional $\mathcal{O}(20)$ integers per event in order to reconstruct the weights. We have derived *analytic weights* for a given set of parameters, based on a detailed analysis of the algorithm, and shown that these *analytic weights* are statistically equivalent at the level of observables to *stochastic weights*, which are derived algorithmically. Finally, we have demonstrated that the implementation of the procedure in the Monte Carlo event generator PYTHIA 8 correctly reweights between sets of flavor selection parameters with a significant timing advantage compared to re-generating samples at new parameter values.

One must, however, keep in mind that the reweighting from an arbitrary point in parameter space to another parameter point may not necessarily be possible—statistical coverage at the new point is required. We demonstrate how a user can test this coverage using the $n_{\text{eff}}/N$ and $1 - \mu$ statistics.

The reweighting methodology developed here has a clear application to tuning—the process of estimating event generator parameters by fitting to data—which is often prohibitively expensive in terms of computational resources, especially when tuning a large number of parameters. In a forthcoming paper, we will demonstrate directly the tuning capabilities of the reweighting procedures introduced here and in ref. [18].

There are, however, a number of other, perhaps less obvious but no less powerful, possible applications for the reweighting methodology developed in this manuscript. One immediate use case is to bias the sampled final state toward rare configurations of interest. For instance, an experiment may wish to study a certain class of baryons for which a dedicated trigger has

been implemented. Without reweighting, one would need to generate events until the particle of interest appears by chance, rejecting the rest, in order to preserve the correct cross section via accept–reject. With hadronization reweighting, one can instead artificially increase the parameters that make the particle more likely to appear, generate the relevant events, and reweight back to the nominal parameter point—automatically recovering the correct cross section. We expect that for many studies at the HL-LHC, such a procedure will not just be preferred, but necessary. Without such methods, the analysis of rare hadronic final states may well become unfeasible due to Monte Carlo statistical uncertainties overwhelming the systematics.

We also foresee applications in model development. It is often desirable to let hadronization parameters depend locally on the properties of the produced hadrons, for instance, their masses or the production location on the string. This creates a chicken-and-egg problem: the parameters one wishes to modify depend on information that is only available after the hadron is produced. One concrete example is the implementation of the rope hadronization model described in ref. [33]. In this model, hadronization parameters depend on the number of strings overlapping with the hadronizing string at the break-up location. To know the location, the hadron mass must be known, and thus its constituent flavors. They are, in turn, affected by the amount of overlap, creating exactly this type of circular dependency. While it is possible to work around this using accept–reject schemes, these often come with high computational cost and only approximate correctness. The ability to use reweighting in cases where a new model can be expressed as a local parameter modification to an existing one is therefore promising, and something we will actively pursue going forward.

Another possible perspective, which may be even further reaching, is the potential for gradient-based tuning of event generators. As shown in ref. [32], since the reweighting scheme provides exact event weights as smooth functions of the hadronization parameters, it becomes possible to compute gradients of observables with respect to those parameters, at least in principle. These gradients differ from traditional finite-difference estimates in that they can be evaluated on a fixed sample without needing to re-generate events. While weight variance and parameter support still introduce noise into these samples, this structure may allow for the use of back-propagation and stochastic gradient descent techniques in localized regions of parameter space. Such an approach could offer an alternative to traditional optimizers, and may open up new ways to interface event generators with machine learning work-flows without resorting to surrogate models. We leave the detailed investigation of this direction to future work.

As a final remark, we would like to ask a question of our readers. The authors of this paper are experienced Monte Carlo developers, and we have seen our share of tricks for variance reduction, fast sampling, and computational efficiency. Yet, when beginning this work, none of us were familiar with the techniques for weight calculation and reweighting that we have described here, and to some extent in our previous paper. A literature search has not revealed them either. While it is entirely possible—even likely—that we have re-discovered techniques well-known in other contexts, the possibility remains that this approach is genuinely new. If so, we note that the substantial speedup in Monte Carlo simulations shown in this paper could be of interest well beyond high-energy physics. Should a reader recognize potential for out-of-field application, we warmly encourage you to get in touch.

## Acknowledgments

We thank Gösta Gustafson, Leif Lönnblad and Torbjörn Sjöstrand for careful reading and constructive comments on the manuscript. CB also thanks GG for patient explanation of Clebsch–Gordan-coefficient derivation.

**Funding information** AY, JZ, MS, and TM acknowledge support in part by the DOE grant de-sc0011784 and NSF OAC-2103889, OAC-2411215, and OAC-2417682. JZ, TM, and MW also acknowledge support in part from the Visiting Scholars Award Program of the Universities Research Association. PI and MW are supported by NSF grants OAC-2103889, OAC-2411215, OAC-2417682, and NSF-PHY-2209769. CB acknowledges support from Vetenskapsrådet contract number 2023-04316 and the LU sustainability fund LU-2023-3031. This document was prepared using the resources of the Fermi National Accelerator Laboratory (Fermilab), a U.S. Department of Energy, Office of Science, Office of High Energy Physics HEP User Facility. Fermilab is managed by FermiForward Discovery Group, LLC, acting under Contract No. 89243024CSC000002.

# A  Extended discussion of analytic and stochastic weights

This appendix serves two purposes: first, to present the full proof of the statistical equivalence between the analytic and stochastic weight prescriptions; and second, to provide a pedagogical introduction to the concepts behind them. We will provide the context in the two first sections, and then go on to the full proof in appendix A.3.

## A.1  Filtered sampling and effective probabilities

The selection of hadrons is subject to additional filters: a hadron is accepted with probability $\epsilon \in [0, 1]$, applied after it has been proposed from a normalized sampling distribution $p$. The sampling process proceeds by proposing candidates, rejecting the proposals with probability $1 - \epsilon$, and continuing until a candidate is accepted.

We consider an observable $\mathcal{O}$, which is a function defined only on the final accepted hadrons. That is, $\mathcal{O}(\text{Accepted})$ has support only on the accepted hadrons, and does not depend on the rejected proposals. The expectation value for the observable under a filtered process is therefore sensitive only to the distribution over accepted candidates, not the details of the rejection chain.

Let us first extend the minimal example for one hadron that we considered in section 2.2, to two hadron types, $h_1$ and $h_2$. These are sampled from a discrete, normalized probability distribution $p(\text{hadron type})$,

$$p(\text{hadron type} = k) = p_1 \delta_{kh_1} + (1 - p_1)\delta_{kh_2}, \tag{A.1}$$

where $\delta_{kh_i}$ is the Kronecker delta, and $p_2 = 1 - p_1$ due to unitarity. Let the filter efficiencies be $\epsilon_1 = 1$, $\epsilon_2 = \epsilon < 1$. The *effective* probability of accepting $h_1$ after any number of rejections is thus

$$p_{1,\text{eff}} = \underbrace{p_1}_{\text{1st attempt}} + \underbrace{p_2(1 - \epsilon_2)p_1}_{\text{2nd attempt}} + \underbrace{p_2^2(1 - \epsilon_2)^2 p_1}_{\text{3rd attempt}} + \dots$$
$$= p_1 \sum_{n=0}^{\infty} (p_2(1 - \epsilon_2))^n = \frac{p_1}{1 - p_2(1 - \epsilon_2)} = \frac{p_1}{p_1 + p_2\epsilon_2}. \tag{A.2}$$

Likewise, the probability $p_{2,\text{eff}}$ of obtaining $h_2$ after the filter is applied, is given by

$$p_{2,\text{eff}} = \underbrace{p_2\epsilon_2}_{\text{1st attempt}} + \underbrace{p_2^2(1 - \epsilon_2)\epsilon_2}_{\text{2nd attempt}} + \underbrace{p_2^3(1 - \epsilon_2)^2\epsilon_2}_{\text{3rd attempt}} + \dots$$
$$= p_2\epsilon_2 \sum_{n=0}^{\infty} (p_2(1 - \epsilon_2))^n = \frac{p_2\epsilon_2}{1 - p_2(1 - \epsilon_2)} = \frac{p_2\epsilon_2}{p_1 + p_2\epsilon_2}. \tag{A.3}$$

This result can be extended to the case of $k = \{1, \ldots, K\}$ hadron types with sampling probabilities $p_k$ and filter efficiencies $\epsilon_k$, where we find that

$$p_{k,\text{eff}} = \frac{p_k \epsilon_k}{\sum_{k'=1}^{K} p_{k'} \epsilon_{k'}} \,. \tag{A.4}$$

These effective probabilities are already correctly normalized, *i.e.*, they satisfy

$$\sum_{k=1}^{K} \frac{\epsilon_k p_k}{\sum_{k'=1}^{K} \epsilon_{k'} p_{k'}} = 1 \,. \tag{A.5}$$

Writing the failed attempts (rejections) as an infinite sum, and identifying that this sum is a geometric series, as in eq. (A.2) and eq. (A.3), underlies the logic of both analytic and stochastic reweighting schemes.

## A.2 Reweighting logic: Analytic and stochastic

The reweighting of a sample obtained using $p_k$, so that it statistically matches a sample generated using an altered probability $p_k'$ (and unchanged filter efficiencies), can be achieved in two ways: either using the stochastic prescription, or the analytic prescription. Before going into the details of the proof of the two prescriptions' equivalence, we give an explanation of the logic.

In the *analytic prescription*, the simulation record consists only of the accepted states after they have passed the filter. The weight for each accepted state $k$ is given by $p_{k,\text{eff}}'/p_{k,\text{eff}}$. In this prescription, the rejection history is encoded in the effective probability by finding the correct expression for the weight analytically, and summing the emerging geometric series, hence the name analytic.

In the *stochastic prescription*, the full rejection history is retained for each accepted state. That is, the simulation record includes both the final accepted candidate $A$, and the sequence of rejected proposals $\{k_1, \ldots, k_{N_R}\}$ that preceded it. Rather than explicitly summing an infinite series over possible rejection histories as in the analytic case, we compute a finite product based on the specific sequence of proposals: each rejected candidate contributes a factor $p_{k,R}'/p_{k,R}$, and the accepted candidate contributes $p_A'/p_A$. The total weight is the product of these ratios. As shown in the following section, these weights will be summed over when computing expectation values over accepted states, thus implicitly summing the infinite series. Since no analytic summation over possible rejection paths is performed, but rather a summation over the actual sampling history, we refer to this as the stochastic prescription.

We now go on to show the general equivalence between the two prescriptions. In section 2.3, we previously illustrated this equivalence in a simple two-type example; here, we give the full general proof.

## A.3 Equivalence of analytic and stochastic weights

We generalize now from the two states $h_1$ and $h_2$, to an arbitrary number of possible states (think about them as possible particle species, *i.e.*, pion, kaon, etc.), which we count with the index $k$. Focusing first on the stochastic prescription, we introduce a bit of notation, summarized in the list below:

- We have $K$ possible states, $k = \{1, \ldots, K\}$, which can be accepted or rejected.

- Each $k$ has a sampling probability $p_k$ with $\sum_{k=1}^{K} p_k = 1$.

- Each $k$ has an associated filter efficiency $\epsilon_k$.

- We denote the number of accepted particles of species $k$ as $n_{A,k}$, and rejected as $n_{R,k}$.

- We use the short-hand $\{n_{A,k}\} \equiv \{n_{A,1}, \ldots, n_{A,K}\}$, to list the number of accepted particles of each species, and similarly $\{n_{R,k}\}$ for rejected.

A simulated event consisting of $N_A$ accepted particles has a *history* of $N_R^n$ rejected particles for each accepted state $n = \{1, \ldots, N_A\}$, with:

$$\sum_k n_{A,k} = N_A, \quad \text{and} \quad \sum_k n_{R,k} = \sum_{n=1}^{N_A} N_R^n = N_R. \tag{A.6}$$

The probability of a given history can then be written as:

$$P_h = \prod_{k=1}^{K} (\epsilon_k p_k)^{n_{A,k}} \left[(1 - \epsilon_k) p_k\right]^{n_{R,k}}. \tag{A.7}$$

The stochastic weight for an event is defined as:

$$w_{\text{stochastic}} = \prod_{k=1}^{K} \left(\frac{p'_k}{p_k}\right)^{n_{A,k} + n_{R,k}}. \tag{A.8}$$

It can easily be checked that the stochastic weight correctly reweights the event's simulation history from $P_h$ to $P'_h$, as:

$$\begin{aligned}
P'_h &= \prod_{k=1}^{K} \left(\epsilon_k p'_k\right)^{n_{A,k}} \left[(1 - \epsilon_k) p'_k\right]^{n_{R,k}} \\
&= w_{\text{stochastic}} P_h.
\end{aligned} \tag{A.9}$$

Note that the stochastic weight makes no reference to efficiencies. They are encoded in the rejection history.

Observable quantities, on the other hand, depend only on the accepted particle species. This means that when the expectation values for the observable quantities are calculated, these are obtained using probabilities $\sum_{\text{Rejected}} P_h$, *i.e.*, summing over all the rejections. In practice this is done automatically, using $P_h$ for each simulation history, but making sure that the samples are large enough so that $\sum_{\text{Rejected}} P_h$ is well approximated. The probability $\sum_{\text{Rejected}} P_h$ equals to

$$\begin{aligned}
\sum_{\text{Rejected}} P_h &= \prod_{n=1}^{N_A} \left( \sum_{N_R^n=0}^{\infty} \prod_{m=1}^{N_R^n} \sum_{k_m=1}^{K} (1 - \epsilon_{k_m}) p_{k_m} \right) \epsilon_n p_n \\
&= \prod_{n=1}^{N_A} \left( \sum_{N_R^n=0}^{\infty} \prod_{m=1}^{N_R^n} \left(1 - \sum_{k=1}^{K} \epsilon_k p_k\right) \right) \epsilon_n p_n \\
&= \prod_{n=1}^{N_A} \left( \sum_{N_R^n=0}^{\infty} \left(1 - \sum_{k=1}^{K} \epsilon_k p_k\right)^{N_R^n} \right) \epsilon_n p_n \\
&= \prod_{n=1}^{N_A} \left( \frac{\epsilon_n p_n}{1 - \left(1 - \sum_{k=1}^{K} \epsilon_k p_k\right)} \right) \\
&= \prod_{k=1}^{K} \left( \frac{\epsilon_k p_k}{\sum_{k'=1}^{K} \epsilon_{k'} p_{k'}} \right)^{n_{A,k}},
\end{aligned} \tag{A.10}$$

where in the last equality we regrouped the probability factors in terms of accepted particle species. Note that $\sum_{\text{Rejected}} P_h$ is given as a product of effective probabilities $\epsilon_k p_k / \sum_{k'=1}^{K} \epsilon_{k'} p_{k'}$ for each accepted particle that enter the definition of the analytic weights,

$$w_{\text{analytic}} = \prod_{k=1}^{K} \left( \frac{p'_k}{\sum_{k'} \epsilon_{k'} p'_{k'}} \Big/ \frac{p_k}{\sum_{k'} \epsilon_{k'} p_{k'}} \right)^{n_{A,k}}. \tag{A.11}$$

We can now show that the two sets of weights are statistically equivalent. That is, the expectation value of any observable $\mathcal{O}$ calculated using the simulation dataset with weights calculated using either of the two prescriptions leads to the same result. Explicitly,

$$
\begin{aligned}
&\mathbb{E}_{\text{Accepted}}\Big[\mathbb{E}_{\text{Rejected}}\big[w_{\text{stochastic}}\mathcal{O}(\{n_{A,k}\})\big]\Big]\\
&= \sum_{\{n_{A,k}\}} \sum_{\{n_{R,k}\}} w_{\text{stochastic}} P_h \mathcal{O}(\{n_{A,k}\})\\
&= \sum_{\{n_{A,k}\}} \sum_{\{n_{R,k}\}} P'_h \mathcal{O}(\{n_{A,k}\})\\
&= \sum_{\{n_{A,k}\}} \prod_{k=1}^{K} \left( \frac{\epsilon_k p'_k}{\sum_{k'=1}^{K} \epsilon_{k'} p'_{k'}} \right)^{n_{A,k}} \mathcal{O}(\{n_{A,k}\})\\
&= \sum_{\{n_{A,k}\}} \prod_{k=1}^{K} \left( \frac{p'_k}{\sum_{k'} \epsilon_{k'} p'_{k'}} \Big/ \frac{p_k}{\sum_{k'} \epsilon_{k'} p_{k'}} \right)^{n_{A,k}} \left( \frac{\epsilon_k p_k}{\sum_{k'=1}^{K} \epsilon_{k'} p_{k'}} \right)^{n_{A,k}} \mathcal{O}(\{n_{A,k}\})\\
&= \mathbb{E}_{\text{Accepted}}\Big[w_{\text{analytic}}(\{n_{A,k}\})\mathcal{O}(\{n_{A,k}\})\Big],
\end{aligned}
\tag{A.12}
$$

where $\mathbb{E}_{\text{Accepted}}$ denotes an average over the accepted particles, and $\mathbb{E}_{\text{Rejected}}$ over rejected particle species. This concludes the proof.

## B  Derivation of the Clebsch–Gordan weights

The Clebsch–Gordan weights in table 1, used to assign filter efficiencies to baryon formation from quark–diquark pairs, are derived from SU(6) spin × flavor symmetry. In this framework, baryons are modeled as bound states of three quarks, with the total wavefunction required to be antisymmetric under exchange of identical fermions. This antisymmetry arises from the combination of a totally antisymmetric color wavefunction, and a spin–flavor wavefunction that is either symmetric (for the decuplet) or of mixed symmetry (for the octet). The Clebsch–Gordan weights are reproduced from ref. [29], for which they were originally derived, but the details of the derivation were omitted. In this appendix, we derive the relevant weights by explicitly constructing these spin–flavor combinations and projecting onto physical baryon states

Starting with the example discussed in section 2.5, we write out the wave functions for the remaining decuplet baryons. With two quarks identical and the third of a different kind, the decuplet becomes (for example):

$$|\Delta^{+}_{\frac{3}{2}\frac{3}{2}}\rangle = (\uparrow\uparrow\uparrow)\frac{1}{\sqrt{3}}(uud + udu + duu), \tag{B.1}$$

$$|\Delta^{+}_{\frac{3}{2}\frac{1}{2}}\rangle = \frac{1}{\sqrt{9}}(\uparrow\uparrow\downarrow + \uparrow\downarrow\uparrow + \downarrow\uparrow\uparrow)(uud + udu + duu). \tag{B.2}$$

The decuplet spin-flavor wave functions are trivially fully symmetric, ensuring compatibility with the antisymmetric color part (given a symmetric spatial part of the wave function, ground state $l = 0$, which is assumed throughout).

For the octet baryons, the spin–flavor wavefunction is of mixed symmetry: symmetric under exchange of two quarks, but antisymmetric with respect to the third. This structure ensures compatibility with the antisymmetric color part of the total wavefunction, while allowing for more diverse spin configurations than in the decuplet.

To construct the mixed symmetry structure, we define the spin–flavor part of the wavefunction as:

$$\psi = \frac{\sqrt{2}}{3}\left(\psi_s^{12}\psi_f^{13} + \psi_s^{13}\psi_f^{12} + \psi_s^{23}\psi_f^{12} - \psi_s^{23}\psi_f^{13} - \psi_s^{13}\psi_f^{23} - \psi_s^{12}\psi_f^{23}\right), \qquad \text{(B.3)}$$

where $\psi_s^{ij}$ and $\psi_f^{ij}$ denote spin and flavor combinations antisymmetric under exchange of quarks $i$ and $j$, respectively. The prefactor $\sqrt{2}/3$ ensures normalization, and follows from requiring the total wavefunction to be normalized when summing over all permutations. To be clear, the three terms (denoted with superscript $12, 13, 23$) are not mutually orthogonal, though they sum to a properly normalized state. For the coming derivation of Clebsch–Gordan weights, this overall normalization factor is important, as it ensures correct relative weights between baryons in the octet and decuplet.

For an explicit example, consider the proton wave function, where the mixed symmetry structure and normalization is evident. Expanding all contributions in terms of symmetrized and antisymmetrized components, we get:

$$|p_{\frac{1}{2}\frac{1}{2}}\rangle = \frac{1}{\sqrt{18}}[2udu(\uparrow\uparrow\downarrow) + 2duu(\downarrow\uparrow\uparrow) + 2uud(\uparrow\uparrow\downarrow)$$
$$- udu(\downarrow\uparrow\uparrow) - duu(\uparrow\downarrow\uparrow) - uud(\uparrow\downarrow\uparrow) - udu(\uparrow\uparrow\downarrow) - uud(\downarrow\uparrow\uparrow) - duu(\uparrow\uparrow\downarrow)]. \qquad \text{(B.4)}$$

Recognizing that these terms naturally group into antisymmetric pairs, we can rewrite the wave function in a more structured form:

$$|p_{\frac{1}{2}\frac{1}{2}}\rangle = \frac{1}{\sqrt{18}}[(udu - duu)(\uparrow\downarrow\uparrow - \downarrow\uparrow\uparrow)$$
$$+ (uud - udu)(\uparrow\uparrow\downarrow - \uparrow\downarrow\uparrow) + (uud - duu)(\uparrow\uparrow\downarrow - \downarrow\uparrow\uparrow)]. \qquad \text{(B.5)}$$

This structured form highlights the underlying antisymmetric structure in both the spin and flavor components, and provides a good strategy for the reader who wishes to write up baryon wave functions for themselves, and check the result. The expression can be constructed directly by writing an antisymmetric piece of the spin wave function as

$$(\uparrow\downarrow - \downarrow\uparrow)\uparrow, \qquad \text{(B.6)}$$

and similarly for the flavor wave function

$$(ud - du)u. \qquad \text{(B.7)}$$

Performing cyclic permutations and normalizing then leads to the final expression in eq. (B.5). In the following, this is the strategy used to write up all the different baryon wave functions.

In the case of the octet, the presence of mixed symmetry required explicit antisymmetrization of the wave function. However, for the decuplet, where all quarks are symmetrized under exchange, we obtain a fully symmetric state with more permutations. For example:

$$|\Sigma_{\frac{3}{2}\frac{3}{2}}^{*0}\rangle = \frac{1}{\sqrt{6}}[(ud + du)s + (ds + sd)u + (us + su)d](\uparrow\uparrow\uparrow), \qquad \text{(B.8)}$$

$$|\Sigma_{\frac{3}{2}\frac{1}{2}}^{*0}\rangle = \frac{1}{\sqrt{18}}[(ud + du)s + (ds + sd)u + (us + su)d](\uparrow\uparrow\downarrow + \uparrow\downarrow\uparrow + \downarrow\uparrow\uparrow). \qquad \text{(B.9)}$$

The octet consists of the two states $\Sigma^0$ and $\Lambda$, which we write as the proton above:

$$|\Sigma^0_{\frac{1}{2}\frac{1}{2}}\rangle = \frac{1}{6}\Big[(\uparrow\downarrow\uparrow - \downarrow\uparrow\uparrow)(usd - sud + dsu - sdu)$$
$$+ (\uparrow\uparrow\downarrow - \uparrow\downarrow\uparrow)(dus - dsu + uds - usd) \tag{B.10}$$
$$+ (\uparrow\uparrow\downarrow - \downarrow\uparrow\uparrow)(uds - sdu + dus - sud)\Big],$$

$$|\Lambda_{\frac{1}{2}\frac{1}{2}}\rangle = \frac{1}{3\sqrt{12}}\Big[(\uparrow\downarrow\uparrow - \downarrow\uparrow\uparrow)(2uds - 2dus + usd - sud - dsu + sdu)$$
$$+ (\uparrow\uparrow\downarrow - \uparrow\downarrow\uparrow)(2sud - 2sdu + dus - dsu - uds + usd) \tag{B.11}$$
$$+ (\uparrow\uparrow\downarrow - \downarrow\uparrow\uparrow)(2usd - 2dsu + uds - sdu - dus + sud)\Big].$$

Similarly for the two remaining diquarks:

$$|(ud)_1\rangle = (\uparrow\uparrow)\frac{1}{\sqrt{2}}(ud + du) = \frac{1}{\sqrt{2}}(u\uparrow d\uparrow + d\uparrow u\uparrow), \tag{B.12}$$

$$|(ud)_0\rangle = \frac{1}{\sqrt{2}}(\uparrow\downarrow - \downarrow\uparrow)\frac{1}{\sqrt{2}}(ud - du) = \frac{1}{2}(u\uparrow d\downarrow - u\downarrow d\uparrow - d\uparrow u\downarrow + d\downarrow u\uparrow). \tag{B.13}$$

To obtain the relative weight for producing a given baryon from a quark–diquark configuration, we project the quark–diquark product state onto the baryon wavefunction, which represents the coupled spin–flavor eigenstate. Concretely, this means calculating the squared amplitude of the overlap between the diquark-quark and the baryon state. It can be useful to think in the more familiar analogy of spin coupling in SU(2) quantum mechanics. Here, we are often faced with the task of having to project a product state, such as $|\uparrow\downarrow\rangle$ (representing two uncoupled spin-1/2 particles, equivalent to our diquark–quark product state) onto a spin eigenstate, such as $|1,0\rangle$ (equivalent to our spin-flavor eigenstate). In this case, the Clebsch–Gordan weight can be calculated as:

$$w = \left|\langle\uparrow\downarrow|\frac{1}{\sqrt{2}}(|\uparrow\downarrow\rangle + |\downarrow\uparrow\rangle)\right|^2 = \frac{1}{2}, \tag{B.14}$$

the standard textbook result, which can be found in any Clebsch–Gordan table.

We can now go on and calculate all the spin × flavor rejection weights in this way. The $(uu)_1 + u$ column of table 1 was already worked out in section 2.5, giving a weight of 4/3. For the purpose of using these weights in a Monte Carlo, having weights larger than unity is impractical, hence we choose this weight as the overall normalization.

For the $(uu)_1 + d$ or $s$ column, we illustrate with $d\downarrow$ ($d\uparrow$ is not allowed) for the octet (getting $p\uparrow$), and get:

$$w_{(uu)_1+d,8} = \left(\frac{1}{\sqrt{18}} + \frac{1}{\sqrt{18}}\right)^2 = \frac{2}{9}, \tag{B.15}$$

which normalized becomes $\boxed{1/6.}$

For the decuplet we illustrate with $d$ (getting $\Delta^+$) and, as before, we get one term for $m = 3/2$ ($d\uparrow$) and one for $m = 1/2$ ($d\downarrow$):

$$w_{(uu)_1+d,10} = \left(\frac{1}{\sqrt{3}}\right)^2 + \left(\frac{1}{\sqrt{9}}\right)^2 = \frac{4}{9}, \tag{B.16}$$

and normalize to get $\boxed{1/3.}$

Next we do the $(ud)_0 + u$ or $d$ column. The decuplet is 0 trivially. For the octet we choose a $u\uparrow$ (getting $p\uparrow$) and get one term for each term in the diquark:

$$w_{(ud)_0 + u\uparrow,8} = \left(\frac{2}{2\sqrt{18}} + \frac{1}{2\sqrt{18}} + \frac{1}{2\sqrt{18}} + \frac{2}{2\sqrt{18}}\right)^2 = \left(\frac{6}{2\sqrt{18}}\right)^2 = \frac{1}{2}. \tag{B.17}$$

The weight for $u\downarrow$ (getting $p\downarrow$) is identical, so the total weight is 1, and we multiply by 3/4 to get $\boxed{3/4.}$

Then the $(ud)_0 + s$ column, where again, the decuplet is 0 trivially. The octet consists of both $\Sigma^0$ and $\Lambda$. We choose $s\uparrow$ (getting $\Sigma^0\uparrow$ and $\Lambda\uparrow$) All terms from $\Sigma^0$ cancel. From $\Lambda$ we get:

$$w_{(ud)_0 + s\uparrow,8} = 0^2 + \left(\frac{1}{2}\frac{1}{3\sqrt{12}}12\right)^2 = \frac{1}{3}. \tag{B.18}$$

The weight for $s\downarrow$ (getting $\Lambda\downarrow$) is identical, so the total weight is 2/3, and we multiply by 3/4 to get $\boxed{1/2.}$

Then we do the $(ud)_1 + u$ or $d$ column. For the octet we choose $u\downarrow$ (getting $p\uparrow$), and again get one term for each term in the diquark:

$$w_{(ud)_1 + u,8} = \left(-\frac{1}{6} + (-)\frac{1}{6}\right)^2 = \frac{1}{9}, \tag{B.19}$$

and multiply by 3/4 to get $\boxed{1/12.}$

For the decuplet we also choose $u$ (getting $\Delta^+$) and get as before one term for $m = 3/2$ (with $u\uparrow$) and one for $m = 1/2$ (with $u\downarrow$):

$$w_{(ud)_1 + u,10} = \left(\frac{1}{\sqrt{6}} + \frac{1}{\sqrt{6}}\right)^2 + \left(\frac{1}{\sqrt{18}} + \frac{1}{\sqrt{18}}\right)^2 = \frac{8}{9}, \tag{B.20}$$

and multiply by 3/4 to get $\boxed{2/3.}$

Finally we do the $(ud)_1 + s$ column. The octet consists of both $\Sigma^0$ and $\Lambda$. All terms from $\Lambda$ cancels, and 4 terms remain from $\Sigma$, giving:

$$w_{(ud)_1 + s,8} = \left(\frac{4}{6\sqrt{2}}\right)^2 + 0^2 = \frac{2}{9}, \tag{B.21}$$

and multiply by 3/4 to get $\boxed{1/6.}$

And for the decuplet we get $\Sigma^{*0}$, as before with one term for $m = 3/2$ (with $s\uparrow$) and one for $m = 1/2$ (with $s\downarrow$):

$$w_{(ud)_1 + s,10} = \left(\frac{1}{\sqrt{2}\sqrt{6}} + \frac{1}{\sqrt{2}\sqrt{6}}\right)^2 + \left(\frac{1}{\sqrt{2}\sqrt{18}} + \frac{1}{\sqrt{2}\sqrt{18}}\right)^2 = \frac{4}{9}, \tag{B.22}$$

and multiply by 3/4 to get $\boxed{1/3.}$

Thus, all the coefficients of table 1 have been reproduced.

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
