# Peer review of "Post-hoc reweighting of hadron production in the Lund string model"

_SciPost Physics, doi:SciPost Phys. 19, 104 (2025)_

## Round 1 · Referee Report · Anonymous (Referee 1) · 2025-6-16

Report

The authors present the first implementation of a reweighting procedure for concrete parameter choices in the Lund string fragmentation model. In particular the approach to store all information required for the subsequent reweighting to any set of parameters is novel and enables post-hoc reweighting. The contribution is timely given future precision experiments that will need more robust uncertainty estimation for all parts of Monte Carlo predictions. This work provides a way to incorporate hadronisation variations into the broader analysis toolchain.

The authors provide a thorough introduction to the model being reweighted and carefully describe the methodology used. The contribution is of high quality, and I strongly recommend publication, given its significant impact, novelty, and overall standard. I list a few minor points of confusion and suggestions below.

Requested changes

  1. In the abstract, the authors claim benefits in terms of "numerical stability." However, as far as I can tell, this phrasing is not used elsewhere in the manuscript and is somewhat confusing. To me, "numerical stability" typically refers to issues arising in e.g. computation scattering amplitudes, where the results become imprecise when approaching certain kinematic limits. I suspect the authors are referring instead to the numerical tests they performed. I recommend clarifying this in the abstract and, if necessary, elsewhere in the manuscript.

  2. The introduction could benefit from additional references. Reweighting is a well-established technique in Monte Carlo event generation, and it might help readers to see this work in the context of related developments. In particular, the authors may wish to consider including references such as arXiv:1310.7439 for NLO subtraction reweighting, arXiv:1605.08256 and arXiv:1606.08753 for parton shower reweighting, and possibly arXiv:1607.00763 for EFT reweighting. There may be other relevant works in the field worth including as well.

  3. In the validation section, the setup for the Pythia runs is, as far as I can tell, not described. This information is important for context and future reproducibility. I would particularly appreciate details about the collider setup used.

  4. This point depends on the response to the previous one. I suspect that the presented validations are for a setup similar to LEP (i.e., electron-positron collisions at relatively moderate centre-of-mass energies). Given the claimed relevance to future LHC analyses, it would be interesting to see validations for such setups as well. While the underlying physics is similar, hadronisation in LHC events involves significantly more particles and would serve as a real stress test of the methodology. I would be curious to see how far the authors can vary their parameters while still recovering the explicit hadronisation parameter variations.

  5. Regarding the validation, the authors only show single-parameter variations. While this already probes all relevant components of the implementation, I would—similar to the point above—suggest also showing joint variations where, for example, all parameters are varied simultaneously. This would provide further insight into the applicability of the method for modern analyses and uncertainty estimation scenarios, which often involve simultaneous parameter variations.

  6. The authors attribute the mismatch in Figure 9 to the lack of support in the reweighting. I am not entirely convinced by this explanation. The corresponding regions do contain events, so the support does not appear to be entirely absent. Shouldn’t the distributions coincide, given sufficient statistics and adequate phase-space overlap? Re-generating Figure 9 with increased statistics could help determine whether the observed mismatch is indeed due to statistical limitations or to insufficient overlap in the hadronisation phase space.

  7. In the conclusion, the authors anticipate potential applications for model development. While I agree that such applications are plausible, the following two sentences are incomprehensible to me. I do not understand how a "chicken-and-egg" problem would arise in this context. This is likely a matter of clarification rather than substance, and I would appreciate it if the authors could elaborate on this point.

Recommendation

Ask for minor revision

  • validity: top
  • significance: top
  • originality: high
  • clarity: top
  • formatting: perfect
  • grammar: perfect

Author:  Christian Bierlich  on 2025-08-14  [id 5725]

(in reply to Report 1 on 2025-06-16)

We thank the referee for the careful reading of our manuscript and for the constructive feedback. Below we address each point in turn.

  1. In the abstract, the authors claim benefits in terms of "numerical stability." However, as far as I can tell, this phrasing is not used elsewhere in the manuscript and is somewhat confusing. To me, "numerical stability" typically refers to issues arising in e.g. computation scattering amplitudes, where the results become imprecise when approaching certain kinematic limits. I suspect the authors are referring instead to the numerical tests they performed. I recommend clarifying this in the abstract and, if necessary, elsewhere in the manuscript.

Our reply: By "numerical stability", we meant that we provided metrics to determine whether the number of generated events is sufficient to reweight to a certain target. The metrics are defined in equations 24 and 25 and illustrated in Table 2. We have adjusted the language in the abstract for clarity, replacing "numerical stability" with "coverage considerations".

  1. The introduction could benefit from additional references. Reweighting is a well-established technique in Monte Carlo event generation, and it might help readers to see this work in the context of related developments. In particular, the authors may wish to consider including references such as arXiv:1310.7439 for NLO subtraction reweighting, arXiv:1605.08256 and arXiv:1606.08753 for parton shower reweighting, and possibly arXiv:1607.00763 for EFT reweighting. There may be other relevant works in the field worth including as well.

Our reply: Thank you for pointing out our oversight. We have included the mentioned references in the introduction.

  1. In the validation section, the setup for the Pythia runs is, as far as I can tell, not described. This information is important for context and future reproducibility. I would particularly appreciate details about the collider setup used.

Our reply: Thank you for bringing this to our attention. We have added the parameters to a footnote in the introduction to section 3.

  1. This point depends on the response to the previous one. I suspect that the presented validations are for a setup similar to LEP (i.e., electron-positron collisions at relatively moderate centre-of-mass energies). Given the claimed relevance to future LHC analyses, it would be interesting to see validations for such setups as well. While the underlying physics is similar, hadronisation in LHC events involves significantly more particles and would serve as a real stress test of the methodology. I would be curious to see how far the authors can vary their parameters while still recovering the explicit hadronisation parameter variations.

Our reply: We appreciate your excitement about this framework! We share it. There are two main hang-ups before the framework would fully work for proton collisions 1) Junctions and associated CG coeffs (for beam remnants) which is not validated and 2) baryon production in the popcorn -- which is the default Pythia model -- which does not yet work. Rather than pushing the framework to break down as is, we would rather improve it until we expect it to produce working weights in pp. We view this paper as a proof of principle, and a coming paper will have a more pragmatic approach. We have clarified this point in the abstract, introduction, and conclusion where we now state "As proof-of-principle, we fully validate this procedure for string systems without junctions or baryon production via the popcorn mechanism." We are working towards validation and implementation with baryon production via junctions and the popcorn mechanism.

  1. Regarding the validation, the authors only show single-parameter variations. While this already probes all relevant components of the implementation, I would—similar to the point above—suggest also showing joint variations where, for example, all parameters are varied simultaneously. This would provide further insight into the applicability of the method for modern analyses and uncertainty estimation scenarios, which often involve simultaneous parameter variations.

Our reply: Figures 9, 10, and 11 and Table 2 show validation results where multiple parameters are varied, where the variation of all parameters simultaneously is addressed in Figure 10 (and Table 2). This is described in the text in the fourth sentence of the fourth paragraph of section 3. We are happy to incorporate any recommended changes to the text to clarify this point or potentially to introduce additional instructive variations.

  1. The authors attribute the mismatch in Figure 9 to the lack of support in the reweighting. I am not entirely convinced by this explanation. The corresponding regions do contain events, so the support does not appear to be entirely absent. Shouldn’t the distributions coincide, given sufficient statistics and adequate phase-space overlap? Re-generating Figure 9 with increased statistics could help determine whether the observed mismatch is indeed due to statistical limitations or to insufficient overlap in the hadronisation phase space.

Our reply: This is correct, but it can be quite difficult to generate sufficient statistics to satisfactorily describe the tails of the distributions. To illustrate, we have repeated the tests with 10,000,000 events and updated the draft accordingly: figures and table 2 have been updated to reflect the increased statistics. As can be seen in figure 9, the mismatch between the tail distributions is improved but not eliminated in this case. We have retained the old figure to illustrate that a factor 10 more statistics only marginally improves the coverage, and added a more detailed discussion, at the end of section 3.1 to clarify this point. We have also retained the old table 2 (now table 3) giving a complete overview of the change of coverage with increased statistics.

  1. In the conclusion, the authors anticipate potential applications for model development. While I agree that such applications are plausible, the following two sentences are incomprehensible to me. I do not understand how a "chicken-and-egg" problem would arise in this context. This is likely a matter of clarification rather than substance, and I would appreciate it if the authors could elaborate on this point.

Our reply: We have elaborated on this point, adding a concrete example with a reference. We hope it is more clear now.

---

## Round 1 · Referee Report · Anonymous (Referee 2) · 2025-6-30

Strengths

  • written very clearly
  • important development for uncertainty estimate
  • timely

Report

The authors present a new algorithm for attaching event-weights to events for hadronic final states in the Monte Carlo event generator Pythia 8 in a new post-hoc scheme. The goal of this approach is to track possible parameter variations that a given final state depends upon. What is usually done at the time of event generation is that a random decision for a given progress in the simulation chain is made and all other possibilities are thrown away. In this approach a given outcome is tracked with a weight information that depends and varies with the parameters that define a given simulation.

This has enormous consequences for the assessment of possible uncertainties for the simulation of the hadronic final state, specifically for the hadronization uncertainties addressed here. The authors demonstrate convincingly that a large number of parameter variations that would normally have to be produced in thousands of individual runs of the simulation are now obtained all at once.

The focus here is on the selection of flavours for the hadrons that result from individual string breaks but there probably is a lot of potential also for application to other parts of the simulation and even way beyond the application in Monte Carlo event generation for hadronic final states but in completely different fields of research.

The paper is written very clearly and the authors give very readable examples on how they implemented the method in specific cases. Furthermore a number of interesting results document the applicability of the approach to the hadronization of the very relevant event generator Pythia 8. The paper appears is timely in the context of the upcoming high luminosity phase of the LHC and addresses the important issue of limitations in computing power for necessary simulations. I therefore highly recommend this manuscript for publication in SciPost.

Requested changes

Two small remarks that might be updated for the published version:

In Fig. 1 the caption and the figure itself are not properly aligned. When the caption refers to up, down, left, right of the figure, it seems that that figure has been rotated.

The parameter xi (concerning diquark selection) is mentioned on p7 but had not been introduced up to this point. Only on p12 is the next mention of this with only a very bried explanation.

Recommendation

Publish (easily meets expectations and criteria for this Journal; among top 50%)

  • validity: top
  • significance: high
  • originality: top
  • clarity: top
  • formatting: excellent
  • grammar: perfect

Author:  Christian Bierlich  on 2025-08-14  [id 5726]

(in reply to Report 2 on 2025-06-30)

We thank the referee for the careful reading of our manuscript and for the constructive feedback. Below we address each point in turn.

In Fig. 1 the caption and the figure itself are not properly aligned. When the caption refers to up, down, left, right of the figure, it seems that that figure has been rotated.

Our reply: Thank you for catching this. We have updated the caption to better reflect the contents of the figure.

The parameter xi (concerning diquark selection) is mentioned on p7 but had not been introduced up to this point. Only on p12 is the next mention of this with only a very bried explanation.

Our reply: We have updated the discussion on pg. 7 to concern the already introduced $\rho$-parameter rather than $\xi$. The logic remains the same.

---

## Editorial Decision

published